# Characterization and regulation of cell cycle-independent noncanonical gene targeting

Shinta Saito [1] & Noritaka Adachi [1] ✉

Homology-dependent targeted DNA integration, generally referred to as gene targeting, provides a powerful tool for precise genome modification; however, its fundamental mechanisms remain poorly understood in human cells. Here we reveal a noncanonical gene targeting mechanism that does not rely on the homologous recombination (HR) protein Rad51. This mechanism is suppressed by Rad52 inhibition, suggesting the involvement of single-strand annealing (SSA). The SSA-mediated gene targeting becomes prominent when DSB repair by HR or end-joining pathways is defective and does not require isogenic DNA, permitting 5% sequence divergence. Intriguingly, loss of Msh2, loss of BLM, and induction of a target-site DNA break all significantly and synergistically enhance SSA-mediated targeted integration. Most notably, SSA-mediated integration is cell cycle-independent, occurring in the G1 phase as well. Our findings provide unequivocal evidence for Rad51-independent targeted integration and unveil multiple mechanisms to regulate SSA-mediated targeted as well as random integration.

Foreign DNA can be integrated into the genome of mammalian cells, albeit at a very low frequency, when a double-strand break (DSB) is present in the genome DNA and the DSB is accidentally repaired using or incorporating the foreign DNA[1]. Unlike lower eukaryotes, in which homology-independent non-targeted integration (i.e., random integration; RI) depends on NHEJ[2], mammalian cells incorporate foreign DNA with no homology to the genome by either of the two different end-joining (EJ) pathways—non-homologous end joining (NHEJ) and DNA polymerase theta-mediated end-joining (TMEJ)—with roughly equal frequency[3]. NHEJ-mediated RI is characterized by ≤2 bp homology at the junctions, whereas TMEJ-mediated RI utilizes 2 to 6 bp of microhomology between foreign DNA and the genome, typically accompanied by large terminal deletions and junctional insertions[3]. When a foreign DNA has regions of homology with the genome, homology-dependent insertions can occur, as has been reported over three decades ago[4,5]; however, this type of homology-mediated insertion (i.e., targeted integration; TI) is even rarer than RI, rendering the gene-targeting technology inefficient and unfeasible for ideal gene therapy[6]. Although the absolute frequency of TI is extremely low, the combined absence of two EJ pathways, NHEJ and TMEJ, results in extraordinarily high efficiency of gene-targeting (i.e., targeted gene replacement) in human somatic cells and mouse ES cells because of rare RI events[3,7]. CRISPR-Cas9-mediated DSB induction at the target locus enhances TI[8], but it is not that the gene-targeting efficiency is dramatically enhanced because RI events are also stimulated at on- and off-target sites[9].

Given the requirement for extensive DNA homology, there seems to be no question as to the crucial contribution of homologous recombination (HR) to TI of a targeting vector, a foreign DNA typically containing two separated regions of homology[1,10,11]. Indeed, TI frequency is markedly reduced when Rad51, an essential HR protein[12], is genetically deleted[10,11,13]. Similar observations have been made in human and mouse cells that are doubly deficient in Rad54 and Rad54B, which both assist in the role of Rad51 to perform the strand exchange reaction during HR[14,15]. Importantly, however, those mutant cells retain the ability to bring about gene targeting although the TI frequency is reduced to less than 5% of that in Rad54/Rad54B-

[1]Department of Life and Environmental System Science, Graduate School of Nanobioscience, Yokohama City University, Yokohama 236-0027, Japan. ✉e-mail: nadachi@yokohama-cu.ac.jp

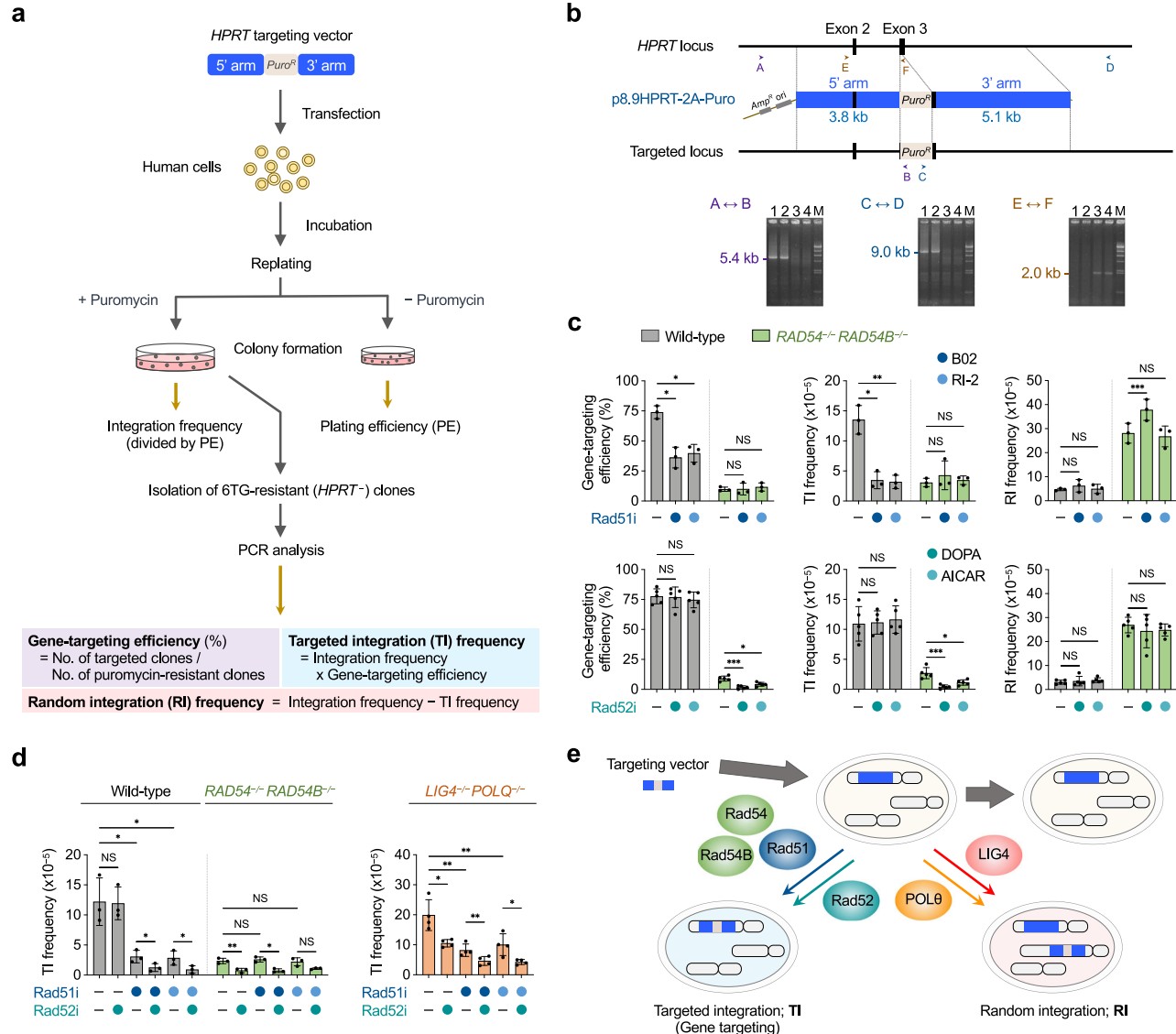

**Fig. 1 | SSA-mediated gene targeting becomes prominent when DSB repair by HR or EJ is deficient. a** Schematic of gene-targeting assay to calculate gene-targeting efficiency, targeted integration (TI) frequency, and random integration (RI) frequency. *Puro*[r], puromycin-resistance gene; 6TG, 6-thioguanine. **b** Scheme of gene targeting assay at the *HPRT* locus. Homologous integration of p8.9HPRT-2A-Puro in 6TG-resistant (i.e., *HPRT*-deficient) cells were confirmed by PCR analysis using three different primer sets (A/B, C/D, and E/F). Shown are the results of three 6TG-resistant clones (lanes 1–3) and a 6TG-sensitive *HPRT*[+] clone (lane 4) (*n* = 1; note that one clone (lane 3) is a random integrant harboring a spontaneous *HPRT* mutation.) M, HindIII-digested λDNA. **c** Impact of Rad51/Rad52 inhibition on gene targeting in wild-type and *RAD54*[−/−]*RAD54B*[−/−] cells. Cells were transfected with linearized p8.9HPRT-2A-Puro, and treated with Rad51 inhibitor (B02 or RI-2) or Rad52 inhibitor (DOPA or AICAR) prior to replating. Data shown are the mean ± s.d. (*n* = 3

for Rad51i; *n* = 5 for Rad52i). **d** Impact of dual inhibition of Rad51 and Rad52 on TI frequency. p8.9HPRT-2A-Puro-transfected cells were treated with either or both of Rad51/Rad52 inhibitors prior to replating. Symbols are as in (**c**). Data shown are the mean ± s.d. (*n* = 3 for wild-type and *RAD54*[−/−]*RAD54B*[−/−]; *n* = 4 for *LIG4*[−/−]*POLQ*[−/−]). Statistical significance in (**c**) and (**d**) was determined by two-sided Student's *t* test. *P < 0.05; **P < 0.01; ***P < 0.001; NS, not significant. Source data are provided as a Source Data file. **e** Schematic representation of targeted and random integration (TI and RI) of targeting vector. RI occurs in a homology-independent manner and relies on either Ku/LIG4-dependent NHEJ or POLθ-dependent TMEJ, while TI is homology-dependent. In addition to HR that relies on Rad51/Rad54/Rad54B, a noncanonical TI mechanism has been uncovered that is dependent on Rad52 but not Rad51. It should be noted that TI and RI are highly infrequent; neither TI nor RI does not occur in >99.99% of transfected cells.

proficient cells[14,15]. This implies the existence of a mechanism responsible for residual gene-targeting events, yet the topic remains an unresolved question. Uncovering this minor pathway is expected not only to improve our understanding of the overall gene-targeting mechanism, but also to elucidate how gene targeting is stimulated by loss of certain DNA repair factors and by DSB induction at the target site. Here, we reveal noncanonical gene targeting that does not rely on Rad51. This mechanism is suppressed by Rad52 inhibition and becomes prominent when other DSB repair pathways are defective. We also find that BLM loss and a target-site DNA break enhance both Rad51-dependent and independent gene targeting.

Strikingly, the occurrence of Rad51-independent gene targeting is not confined to the S-G2 phase of the cell cycle.

## Results

### Genetic evidence of a Rad51-independent mechanism for gene targeting

To investigate the mechanism of TI events occurring in HR-compromised cells, we used Rad54/Rad54B double-knockout (*RAD54*[−/−]*RAD54B*[−/−]) human Nalm-6 cells[15] to examine the effect of Rad51 inhibition on gene targeting at the *HPRT* locus (Fig. 1a) using a promoterless-type vector with 8.9-kb homology arms (a 3.8-kb 5′ arm

and a 5.1-kb 3' arm) as shown in Fig. 1a. Two different chemical inhibitors, B02[16] and RI-2[17], were employed for Rad51 inhibition and, as expected, both inhibitors significantly reduced TI frequency in wild-type Nalm-6 cells, but had no impact on TI frequency in $RAD54^{-/-}$ $RAD54B^{-/-}$ cells (Fig. 1c), suggesting that cells lacking Rad54 and Rad54B are defective in Rad51-dependent HR and thus inert to cause HR-mediated TI.

## SSA is responsible for Rad51-independent gene targeting

We then investigated the impact of Rad52 inhibition on gene targeting. Unlike yeast Rad52, which plays critical roles in HR as well as single-strand annealing (SSA)[11], mammalian Rad52 has a limited function during the Rad51-dependent HR reaction[18], but plays an important role in DSB repair by SSA, a mechanism that utilizes short (typically >20 bp) homologous sequences present in the vicinity of a DSB[19,20]. Given that the above-mentioned Rad51-independent mechanism relies on SSA, we reasoned that Rad52 inhibition would result in a decreased gene-targeting efficiency in $RAD54^{-/-}RAD54B^{-/-}$ cells, but not in HR-normal cells. This was indeed the case: TI frequency was decreased 2- to 7-fold in $RAD54^{-/-}RAD54B^{-/-}$ cells when treated with Rad52 inhibitor (6-hydroxy-DL-DOPA (DOPA)[21] or AICAR[22]) (Fig. 1c). Decreased TI frequencies were also observed in shRad52-transfected $RAD54^{-/-}$ $RAD54B^{-/-}$ cells (Supplementary Fig. 1a,b). In sharp contrast, Rad52 inhibition and shRad52 transfection had no impact on TI frequency in wild-type cells (Fig. 1c and Supplementary Fig. 1a,b). These data support the view that Rad52 is involved in a Rad51/Rad54/Rad54B-independent route for gene targeting. Consistent with this, inhibition of both Rad51 and Rad52 in wild-type cells had a stronger effect on TI suppression than did Rad51 inhibition alone, whereas in $RAD54^{-/-}$ $RAD54B^{-/-}$ cells, the degree of TI reduction caused by the dual inhibition was essentially the same as that by Rad52 inhibition alone (Fig. 1d and Supplementary Fig. 1c). Together, these results indicate that Rad52 is involved in gene targeting in $RAD54^{-/-}RAD54B^{-/-}$ cells or Rad51-inhibited cells, thus suggesting a role for SSA in rare TI events occurring in HR-compromised cells (Fig. 1e).

To confirm the HR-deficient status of $RAD54^{-/-}RAD54B^{-/-}$ cells, we performed the DR-GFP reporter assay, which has been widely used to evaluate HR activity[23]. Whereas wild-type cells displayed an ~70-fold increased HR frequency upon I-SceI-mediated DSB induction in a Rad51-dependent manner, such a marked enhancement of HR was not observed in $RAD54^{-/-}RAD54B^{-/-}$ cells (Supplementary Fig. 2a,b). We then performed the SA-GFP reporter assay[24] to assess the SSA status of $RAD54^{-/-}RAD54B^{-/-}$ cells. We found that $RAD54^{-/-}RAD54B^{-/-}$ cells exhibit 3-fold enhanced SSA frequencies as compared with wild-type cells (Supplementary Fig. 2d–f). Rad52 inhibition resulted in a 2 to 3-fold decreased SSA frequency in $RAD54^{-/-}RAD54B^{-/-}$ cells (Supplementary Fig. 2f), but did not influence the HR frequency, as determined by the DR-GFP reporter assay (Supplementary Fig. 2c), again confirming a limited contribution of Rad52, if any, to HR (Rad51)-mediated DSB repair and gene targeting. It is interesting to note that SSA appears to occur more rapidly than HR in chromosomal DSB repair of human cells, given our results of DR-GFP and SA-GFP assays (Supplementary Fig. 2g).

Although our results point to the existence of an SSA-mediated mechanism for TI in human cells, it remained unclear whether this noncanonical mechanism only operates when HR is compromised. Previous work has shown that loss of DNA ligase IV (LIG4; essential for NHEJ) leads to increased HR frequencies[3]. Likewise, DNA polymerase theta (POLθ; POLQ), which is essential for TMEJ[3,7], is suggested to suppress HR through Rad51 inhibition[25]. Therefore, we next employed EJ-deficient LIG4/POLQ double-knockout ($LIG4^{-/-}POLQ^{-/-}$) Nalm-6 cells, where homology-independent RI does not occur[3] (cf. Fig. 1e). The DR-GFP and SA-GFP reporter assays showed that HR and SSA are both enhanced in $LIG4^{-/-}POLQ^{-/-}$ cells (Supplementary Fig. 2c,f). These enhancements were largely attributed to LIG4 deficiency, as TMEJ loss

only had a marginal effect when NHEJ is active (Supplementary Fig. 2h). As shown in Fig. 1d, gene-targeting assays revealed that $LIG4^{-/-}POLQ^{-/-}$ cells exhibited 1.5-2-fold higher TI frequencies than did wild-type cells. Interestingly, in $LIG4^{-/-}POLQ^{-/-}$ cells, inhibition of Rad51 and Rad52 similarly reduced TI frequency, and the dual inhibition showed an additive effect on suppressing TI (Fig. 1d). These results show that elimination of EJ pathways promotes not only HR-dependent TI but also TI through an SSA-mediated mechanism. Hereafter, HR-dependent TI (which is Rad51-dependent) will be referred to as "HR-TI", and HR-independent TI (which is Rad51-independent and mediated by SSA) as "SSA-TI".

## SSA-based gene targeting is tolerant to sequence divergence

We next sought to determine whether SSA-TI requires perfect sequence homology between the arms of a targeting vector and the genome. In the yeast *Saccharomyces cerevisiae*, repair by SSA can occur even between divergent sequences, particularly when cells are deficient in the mismatch repair protein Msh2, which has been implicated in heteroduplex rejection during DNA recombination[26,27]. We therefore predicted that SSA-TI might not require isogenic arms with sequences identical to the genome, and that the frequency of SSA-TI could be negatively affected by Msh2, given its anti-recombination function as mentioned above. To test these predictions, we set out to prepare two necessary items. Firstly, we constructed a series of targeting vectors harboring different percentages of mutations (base substitutions) in either or both arms (Fig. 2a). Specifically, these vectors contain a 1.7-kb 5' arm with 100% or 95% homology to the genome (*HPRT* intron 2) and a 1.3-kb 3' arm with 100%, 99%, 97.5%, 95%, 90%, or 80% homology to the genome (*HPRT* intron 3) (Fig. 2a and Supplementary Methods). These mutations were introduced at equal intervals in intronic regions (other than splice donor/acceptor sites) in a way that does not change overall GC content of the arm DNA. Secondly, we generated a set of cell lines proficient or deficient in Msh2 for each genotype (i.e., wild-type, $RAD54^{-/-}RAD54B^{-/-}$, and $LIG4^{-/-}POLQ^{-/-}$ cells). Since Nalm-6 cells lack Msh2 expression due to a large deletion at the *MSH2* loci[28], we restored Msh2 expression by knocking in a super-exon of *MSH2* (Supplementary Fig. 3a). The recovery of Msh2 function in each cell line was verified by restoration of Msh6 as well as Msh2 expression (Supplementary Fig. 3b). Using the systematically constructed targeting vectors and Msh2-proficient/deficient cell lines, we examined the impact of arm sequence divergence and cellular Msh2 status on gene targeting.

Overall, the results obtained using targeting vectors with a total arm length of 3.0 kb showed that as arm sequence homology is decreased, TI frequency is reduced (Fig. 2b and Supplementary Fig. 4). This tendency was evident in all cell lines used and appeared more prominent in Msh2-proficient cells. Importantly, given that $RAD54^{-/-}RAD54B^{-/-}$ cells are deficient in HR-TI, it is obvious that Msh2 acts to inhibit SSA-TI: in Msh2-proficient $RAD54^{-/-}RAD54B^{-/-}$ cells, SSA-TI was abolished by the presence of 5% sequence divergence in one arm (1 mutation every 20 bp) (Fig. 2b). In the absence of Msh2, SSA-TI is markedly enhanced, but no longer occurs when the divergence in one arm exceeds 20%. Thus, SSA-TI does not require perfect sequence homology between vector arms and the genome, but is affected by sequence divergence and is significantly suppressed by Msh2.

Not surprisingly, HR-TI is more severely affected by arm sequence divergence (Fig. 2b). When 1% or 2.5% divergence was present in the 3' arm (1 mutation every 100 or 40 bp, respectively), the frequency of TI was reduced or nearly completely abolished in Msh2-proficient cells; however, such reduction was not seen in Msh2-deficient $RAD54^{-/-}$ $RAD54B^{-/-}$ cells, suggesting that only a 1% divergence in one arm decreases HR-TI (Fig. 2b and Supplementary Fig. 4b). Notably, 5% sequence divergence in both arms abolishes HR-TI because it reduced the frequency of TI in wild-type cells to the level comparable to that in $RAD54^{-/-}RAD54B^{-/-}$ cells (Fig. 2b,c, and Supplementary Fig. 4a,b); namely, the TI frequency of a targeting vector with 5% divergence (in

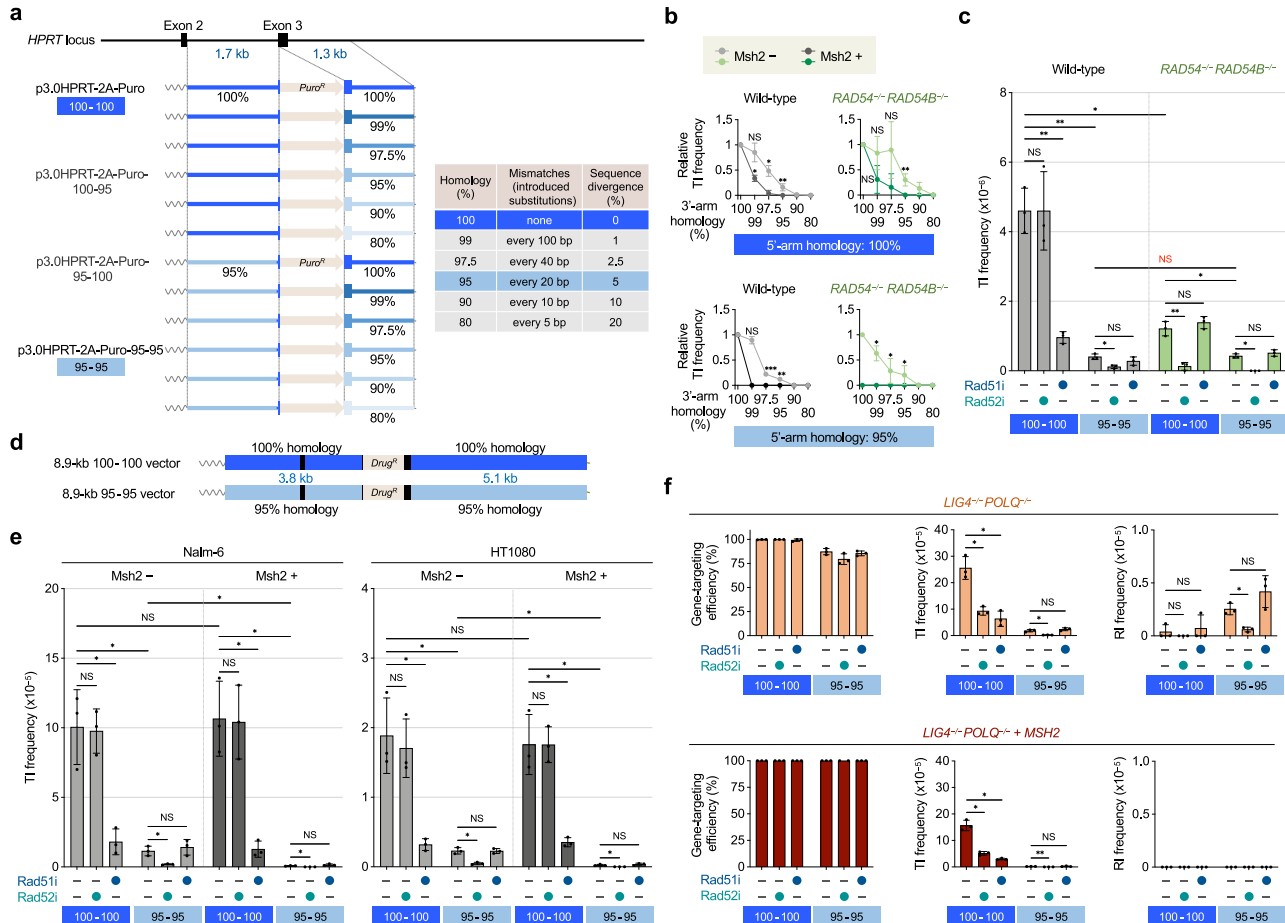

**Fig. 2 | SSA-mediated DNA integration is tolerant to sequence divergence and is suppressed by Msh2. a** Schematic representation of *HPRT* targeting vectors with different arm sequence homology. Shown are 12 targeting vectors with a total arm length of 3.0 kb containing a 1.7-kb 5′ arm with 100% or 95% homology to the genome (intron 2) and a 1.3-kb 3′ arm with 100%, 99%, 97.5%, 95%, 90%, or 80% homology to the genome (intron 3). The wavy lines indicate the plasmid backbone. **b** Impact of arm sequence divergence on TI frequency. Msh2-deficient (-) and proficient (+) Nalm-6 cells were transfected with vectors shown in (**a**). Each graph shows TI frequencies of divergent vectors relative to the 100-100 vector p3.0HPRT-2A-Puro (top) or the 95-100 vector (bottom). Where absent, error bars fall within symbols. **c** Impact of Msh2 deficiency on TI frequency in wild-type (WT) and *RAD54⁻/⁻ RAD54B⁻/⁻* cells. After transfection of linearized p3.0HPRT-2A-Puro (100-

100) or p3.0HPRT-2A-Puro-95-95 (95-95), cells were treated with DOPA or B02 prior to replating. Symbols are as in Fig. 1c. Data shown are the mean ± s.d. (*n* = 3). **d** Schematic representation of *HPRT* targeting vectors with a total arm length of 8.9 kb. Symbols are as in (**a**). **e** Impact of Msh2 deficiency on TI frequency. Msh2-deficient (-) and proficient (+) cell lines from Nalm-6 and HT1080 were transfected with vectors shown in **d**, and treated with DOPA or B02 prior to replating. Symbols are as in Fig. 1c. Data shown are the mean ± s.d. (*n* = 3). **f** Impact of Msh2 deficiency on gene targeting in *LIG4⁻/⁻ POLQ⁻/⁻* cells. p8.9HPRT-2A-Puro or p8.9HPRT-2A-Puro-95-95 transfected cells were treated with DOPA or B02 prior to replating. Symbols are as in Fig. 1c. Data shown are the mean ± s.d. (*n* = 3). Statistical significance in (**b, c, e,** and **f**) was determined by two-sided Student's *t* test. *P < 0.05; **P < 0.01; ***P < 0.001; NS, not significant. Source data are provided as a Source Data file.

which both arms are 95% homologous to the genome; hereafter referred to as the "95-95" vector) was ~25-fold lower than that of the original vector ("100-100") in wild-type cells and was essentially the same as that seen in *RAD54⁻/⁻ RAD54B⁻/⁻* cells (Fig. 2b,c, and Supplementary Fig. 4a,b). It is interesting to note that this finding also reveals the fact that SSA-TI does occur, albeit rarely, when HR is proficient.

Consistent with the observation that HR and SSA are both enhanced in cells deficient in LIG4 and POLQ, TI frequencies in these cells were significantly higher than in wild-type cells, irrespective of arm sequence divergence or Msh2 status. Of note, the 95-95 vector showed a three-fold higher TI frequency in *LIG4⁻/⁻ POLQ⁻/⁻* cells than in wild-type and *RAD54⁻/⁻ RAD54B⁻/⁻* cells (Supplementary Fig. 4c), again confirming that SSA-TI frequency is elevated when EJ pathways are eliminated.

To confirm our observations made with the 3.0 kb-arm targeting vectors, we next constructed an 8.9 kb-arm "95-95" vector (Fig. 2d and Supplementary Methods) to further investigate the mechanism and property of SSA-TI. As expected, TI frequency of the 95-95 vector was ~10-fold lower than that of the 100-100 vector (i.e., p8.9HPRT-2A-Puro)

and was further 15-fold lower when Msh2 was present (Fig. 2e and Supplementary Fig. 5). Consistent with the observations shown in Fig. 1c, the TI frequency of the 100-100 vector was reduced by Rad51 inhibition (~5-fold), and not by Rad52 inhibition. In stark contrast, the TI frequency of the 95-95 vector was unaffected by Rad51 inhibition, but was markedly (~6-fold) reduced by Rad52 inhibition (Fig. 2e and Supplementary Fig. 5). Opposing effects by Rad51 and Rad52 inhibitors were similarly observed using the 3.0 kb-arm vectors (Fig. 2c). Additionally, consistent with the results obtained with Rad52 inhibitor, Rad52 knockout *RAD52⁻/⁻* cells (which exhibit reduced capacity for SSA, not HR, as revealed by DR-GFP and SA-GFP reporter assays; Supplementary Fig. 6a–d) displayed ~15-fold reduced TI frequencies for the 95-95 vector (but not for the 100-100 vector) particularly when Msh2 is absent (Supplementary Fig. 6e). Importantly, this SSA-TI reduction in 95-95 vector-transfected *RAD52⁻/⁻* cells was not further reduced by either Rad51 or Rad52 inhibition (Supplementary Fig. 6e). We also note that similar to Rad52 inhibition or genetic deletion, SSA-TI reduction with the 95-95 vector was observed in gene-knockdown experiments using siRNA for XPF/ERCC1, which are both implicated in SSA[29]

(Supplementary Fig. 6f, g). Together, our findings reveal that targeting vectors with 5% arm sequence divergence do not integrate into the genome via HR, but can cause SSA-mediated TI, albeit rarely, even when HR is functionally normal.

In the HR-deficient $RAD54^{-/-}RAD54B^{-/-}$ cell line, the 5% divergence in the arms only resulted in a slight decrease in TI frequency (~2-fold) (Fig. 2c and Supplementary Fig. 5). For both 100-100 and 95-95 vectors, Rad51 inhibition had little or no influence, whereas Rad52 inhibition significantly reduced the TI frequency (Fig. 2c and Supplementary Fig. 5). These results are consistent with the notion that $RAD54^{-/-}$ $RAD54B^{-/-}$ cells rely on SSA to cause TI. Even more interesting results were obtained with $LIG4^{-/-}POLQ^{-/-}$ cells, as these cells are highly competent in HR-TI and SSA-TI. For the 95-95 vector, Rad52 inhibition and Msh2 expression synergistically reduced the TI frequency, whereas Rad51 inhibition did not (Fig. 2f). For the 100-100 vector, by contrast, Rad52 inhibition as well as Rad51 inhibition conferred a 3 to 4-fold decrease in TI frequency. Furthermore, unlike in wild-type cells, Msh2 expression in $LIG4^{-/-}POLQ^{-/-}$ cells resulted in an ~2-fold decreased TI frequency (Fig. 2f), again confirming the frequent occurrence of SSA-TI in EJ-deficient cells, especially when a targeting vector with 100% homologous arms is used.

To verify the generality of SSA-TI, we next employed another cell line, HT1080, which has been preferentially used for mechanistic analysis of human cell gene targeting[6,30,31]. We observed that gene targeting did occur in HT1080 cells when the 95-95 vector is used, although it was ~100-fold less efficient than the 100-100 vector (Fig. 2e). $MSH2$ gene-knockout in HT1080 cells (Supplementary Fig. 3c, d) did not affect TI frequency of the 100-100 vector, but markedly (~10-fold) increased TI frequency of the 95-95 vector (Fig. 2e). Furthermore, the opposing effects by Rad51/Rad52 inhibition on TI frequency were observed in HT1080 and its $MSH2^{-/-}$ cells, as is also observed in Nalm-6 cells (Fig. 2e). Collectively, our findings demonstrate that SSA-TI is a general phenomenon, albeit less frequent than HR-TI.

The generality of SSA-TI events for gene targeting led us to consider the necessity of revisiting rare RI clones obtained from $LIG4^{-/-}POLQ^{-/-}$ cells[3]. Specifically, since Nalm-6 is Msh2 defective, we reasoned that those RI events, mediated by homeologous recombination between two Alu sequences, were mechanistically reliant on SSA and hence could be abolished when Msh2 is expressed. This was indeed the case: in $LIG4^{-/-}POLQ^{-/-}$ cells, RI of the 100-100 vector was all dependent on Alu-mediated recombination between the vector and the genome (Fig. 2f and Supplementary Fig. 7a, b); however, these Alu-mediated RI events were never observed in Msh2-proficient cells and even in Msh2-deficient cells treated with Rad52 inhibitor (Fig. 2f). It is particularly interesting to note that when Msh2 is absent, the 95-95 vector displayed significantly (~7 fold) higher RI frequencies than the 100-100 vector (Fig. 2f). It may be that HR-TI incompetent targeting vectors may provoke SSA-mediated RI events ("SSA-RI") more frequently than isogenic targeting vectors.

## Targeted DSB induction enhances SSA-TI

Pioneering work by ref. 32 has demonstrated that induction of a DSB at the target locus significantly enhances gene targeting. Nowadays, DSB induction at a desired endogenous locus can easily be performed with the use of artificial nucleases as exemplified by CRISPR/Cas9[33] and indeed is shown to enhance gene targeting[8]; however, the precise mechanism underlying Cas9 DSB-induced enhancement of TI is unclear. We therefore sought to elucidate whether Cas9-associated TI events rely on SSA in addition to HR.

Intriguingly, Cas9 DSB induction in Nalm-6 cells increased TI frequency 6-fold for the 100-100 vector and 15-fold for the 95-95 vector (Fig. 3a,b and Supplementary Fig. 8a), implying a preferential stimulation of SSA-TI upon DSB induction. Similar but less pronounced results were obtained in $LIG4^{-/-}POLQ^{-/-}$ cells; in contrast, $RAD54^{-/-}$

$RAD54B^{-/-}$ cells exhibited an ~15-fold increased TI frequency upon DSB induction for both 100-100 and 95-95 vectors (Fig. 3a,b and Supplementary Fig. 8a). These results suggest that DSB induction at the target locus stimulates SSA-TI more strongly than HR-TI. This notion is further confirmed by subsequent experiments using Rad51 and Rad52 inhibitors. The TI frequency of the 95-95 vector upon DSB induction was little affected by Rad51 inhibition but was ~6 fold reduced by Rad52 inhibition (Fig. 3c and Supplementary Fig. 8b). Surprisingly, Cas9 DSB-enhanced TI frequency of the 100-100 vector was not only reduced by Rad51 inhibition but also by Rad52 inhibition, as had not been observed in spontaneous TI events (Fig. 2c,e). Thus, our results suggest that targeted DSB induction stimulates SSA-TI even when the vector DNA is competent at causing HR-TI. Absence of Msh2 enhanced Cas9 DSB-induced SSA-TI in all cell lines, but the impact of Rad51 or Rad52 inhibition on TI was essentially the same, regardless of arm sequence divergence of the targeting vector (Fig. 3c and Supplementary Fig. 8b).

The existence of SSA-TI as an additional mechanism for gene targeting in human cells and its preferential stimulation by DSB induction prompted us to investigate whether SSA-TI can occur with short targeting vectors with a total arm length of 40, 80, or 212 bp (Supplementary Fig. 8c). These three vectors were incompetent at causing gene targeting in wild-type cells. In $LIG4^{-/-}POLQ^{-/-}$ cells, the 40-bp arm vector was unable to cause TI, but the 80-bp and 212-bp arm vectors did cause TI, particularly when a targeted DSB was introduced (Supplementary Fig. 8d). Interestingly, TI frequency of the 212-bp vector was comparable to that of the 3.0-kb 95-95 vector (cf. Supplementary Figs. 4c and 8d). As expected, TI of these short-arm vectors in $LIG4^{-/-}POLQ^{-/-}$ cells was strongly suppressed by Msh2, and affected by inhibition of Rad52 (and not Rad51) (Supplementary Fig. 8d). These results suggest that in EJ-deficient cells, SSA-TI permits gene targeting even when a total arm length of the vector is as short as ~80 bp. It is tempting to argue that this situation of human cells is reminiscent of yeast cells in that DSB repair does not rely on EJ and that gene targeting can occur even with short-arm vectors and independently of Rad51[34,35].

## G1-phase DSB induction strongly enhances SSA-TI

We next set out to examine whether SSA-TI occurs in S-G2 phase of the cell cycle as does HR-TI or whether SSA-TI occurrence is cell cycle-independent. Numerous reports have suggested that SSA operates in the S-G2 phase, partly because SSA, like HR, mechanistically requires end resection to generate long 3′-ssDNA tails[36]. It was recently reported, however, that end resection can be observed in G1 or G0 phase as well[37,38]. Additionally, in earlier studies using yeast, Rad52 was shown to compete with Ku (a heterodimer of Ku70 and Ku80 that is absolutely required for NHEJ) at DSB ends[39]. Given our observation that loss of NHEJ enhances SSA more significantly than HR, we speculated that SSA could also occur in the G1 phase.

In order to investigate the cell cycle dependence of SSA-TI, we sought to develop a system to induce a DSB in a cell cycle-dependent manner. For this purpose, we constructed several vectors that, based on the fluorescent ubiquitination-based cell cycle indicator (Fucci) system[40], allow for G1- or S/G2-specific expression of Cas9. In these vectors, Cas9 cDNA is fused to a partial cDNA of human Geminin or Cdt1, as depicted in Fig. 3d. In the Fucci system, which enables the visualization of cell cycle progression in living cells[40], addition of N-terminal 110 amino acids of human Geminin (hGeminin(1/110)) to a fluorescent protein results in expression in the S/G2/M phases (i.e., outside G1 phase), while addition of N-terminal 100 amino acids of human Cdt1 (hCdt1/(1/100)) results in G1-specific expression[40] (Fig. 3d). We also generated vectors expressing Cas9 fused to hCdt1(30/120), which allows for expression in G1 plus early-S phase, or hCdt1(1/100)Cy(-), which allows for expression outside S phase[40]. Transient transfection of these vectors showed that expression levels of Cas9-fusion proteins were lower than Cas9 (Supplementary Fig. 9a), consistent with expectant cell cycle-regulated degradation of these

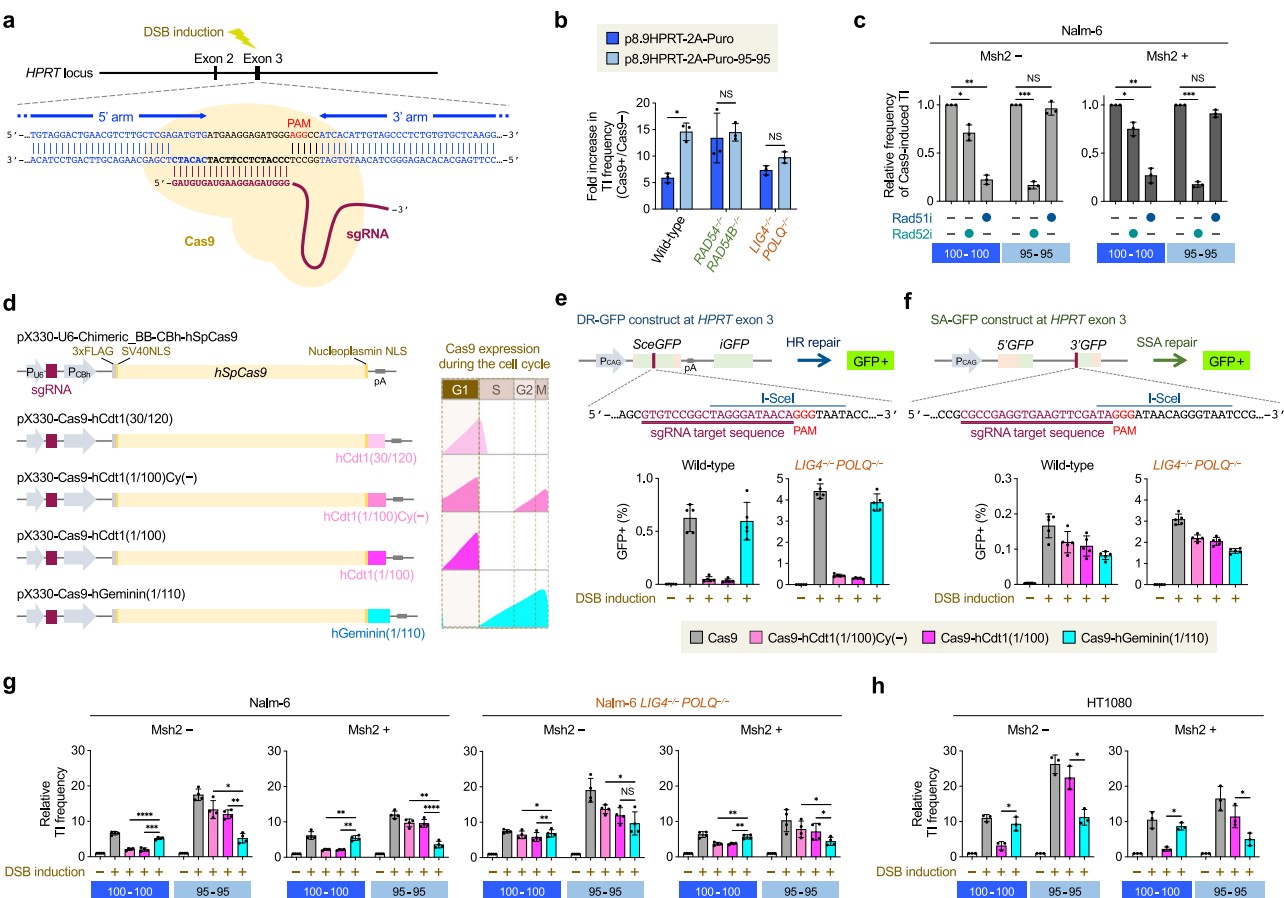

**Fig. 3 | G1-phase DSB induction enhances SSA-mediated gene targeting.**
**a** Schematic diagram of Cas9-mediated DNA cleavage at the *HPRT* gene using exon 3-targeting single-guide RNA (sgRNA). PAM, protospacer-adjacent motif. **b** Impact of Cas9-induced cleavage on TI frequency. Cells were transfected with p8.9HPRT-2A-Puro or p8.9HPRT-2A-Puro-95-95 along with pX330-Cas9-HPRT-Ex3 or pX330-U6-Chimeric_BB-CBh-hSpCas9. The ratio of TI frequency with DSB induction (Cas9+) to that without induction (Cas9-) is indicated. Data shown are the mean ± s.d. (*n* = 3). **c** Impact of Rad51 and Rad52 inhibition on Cas9-induced TI. Msh2-deficient (-) and proficient (+) Nalm-6 cell lines were subjected to co-transfection of targeting vector and pX330-Cas9-HPRT-Ex3, and treated with DOPA or B02 prior to replating. Data shown are the mean ± s.d. (*n* = 3). **d** Schematic representation of cell cycle-dependent Cas9 expression vectors. Expression patterns of each Cas9 protein inferred from the Fucci system are also indicated. P_CBh, CMV enhancer/chicken β-actin hybrid promoter; NLS, nuclear localization signal. **e** Impact of cell cycle-

regulated Cas9 expression on chromosomal HR repair. The sequence of *SceGFP*-targeting sgRNA is indicated, along with PAM and the I-SceI site. Data shown are the mean ± s.d. (*n* = 5). **f** Impact of cell cycle-regulated Cas9 expression on chromosomal SSA repair. The sequence of *3'GFP*-targeting sgRNA is indicated, along with PAM and the I-SceI site. Data shown are the mean ± s.d. (*n* = 5). **g** Impact of targeted DSB induction on TI frequency. The TI frequency of p8.9HPRT-2A-Puro or p8.9HPRT-2A-Puro-95-95 without Cas9 in each cell line was taken as 1, and the relative TI frequency was calculated. Data shown are the mean ± s.d. (*n* = 4).
**h** Impact of targeted DSB induction on TI frequency in HT1080 cells. The TI frequency of p8.9HPRT-2A-Bsr or p8.9HPRT-2A-Bsr-95-95 without Cas9 was taken as 1 in each cell line, and the relative TI frequency was calculated. Data shown are the mean ± s.d. (*n* = 3). Statistical significance in (**b**, **c**, **g**, and **h**) was determined by two-sided Student's *t* test. *P < 0.05; **P < 0.01; ***P < 0.001; ****P < 0.0001; NS, not significant. Source data are provided as a Source Data file.

proteins. Moreover, immunoblot analysis after cell sorting indicated that Cas9-hGeminin(1/110) was expressed outside G1 but was undetectable in G1, whereas Cas9-hCdt1/(1/100) was highly expressed in G1 and was hardly detectable outside G1 (Supplementary Fig. 9b-d). Confirming the cell cycle-regulated expression of Cas9, we performed a modified DR-GFP assay, in which the target substrate (i.e., *SceGFP*) was cleaved with Cas9 instead of I-SceI (Fig. 3e). Expression of Cas9-hGeminin(1/110) markedly increased the frequency of HR, as compared with Cdt1-fused Cas9 proteins (Fig. 3e and Supplementary Fig. 9e). These results confirmed that Cas9-hGeminin(1/110) induces DSBs in the S-G2 phase to stimulate HR, whereas Cas9-hCdt1 proteins only induce DSBs that are ~15-fold less efficient than Cas9 or Cas9-hGeminin(1/110) in enhancing HR. We next performed a modified SA-GFP assay using the expression vectors for Cas9-hGeminin(1/110) (hereafter "S/G2-Cas9") and Cas9-hCdt1 proteins (hereafter "G1-Cas9"). Intriguingly, expression of G1-Cas9 showed a significant effect in enhancing SSA, suggesting that SSA-mediated DSB repair operates in G1 as well as in the S-G2 phase (Fig. 3f and Supplementary Fig. 9f). One

might think that G1 occurrence of SSA needs to be interpreted with caution because Cas9-induced DSBs are unique in that they are RNA-mediated; in addition, G1-Cas9 protein could possibly persist at the break site until the S-G2 phase to cause SSA[41–43]. In this respect, it is important to mention that G1-Cas9 expression is highly inefficient in enhancing HR (Fig. 3e and Supplementary Fig. 9e). Moreover, we obtained essentially the same results with I-SceI endonuclease; specifically, I-SceI-hCdt1 does enhance SSA similarly to I-SceI-hGeminin (Supplementary Fig. 9g). Thus, these results suggest that SSA enhancement upon G1-phase DSBs is a general phenomenon, not confined to Cas9-induced DSBs.

We then proceeded to the gene-targeting assay to analyze the cell cycle dependence of SSA-TI and HR-TI. TI frequency of the 100-100 vector was significantly (>5-fold) increased when Cas9 or S/G2-Cas9 expression vector was co-transfected into Nalm-6 cells, whereas such increase was not seen with co-transfection of G1-Cas9 vectors, and this tendency was similarly observed in Msh2-proficient cells (Fig. 3g and Supplementary Fig. 9h). These results are in perfect accordance with

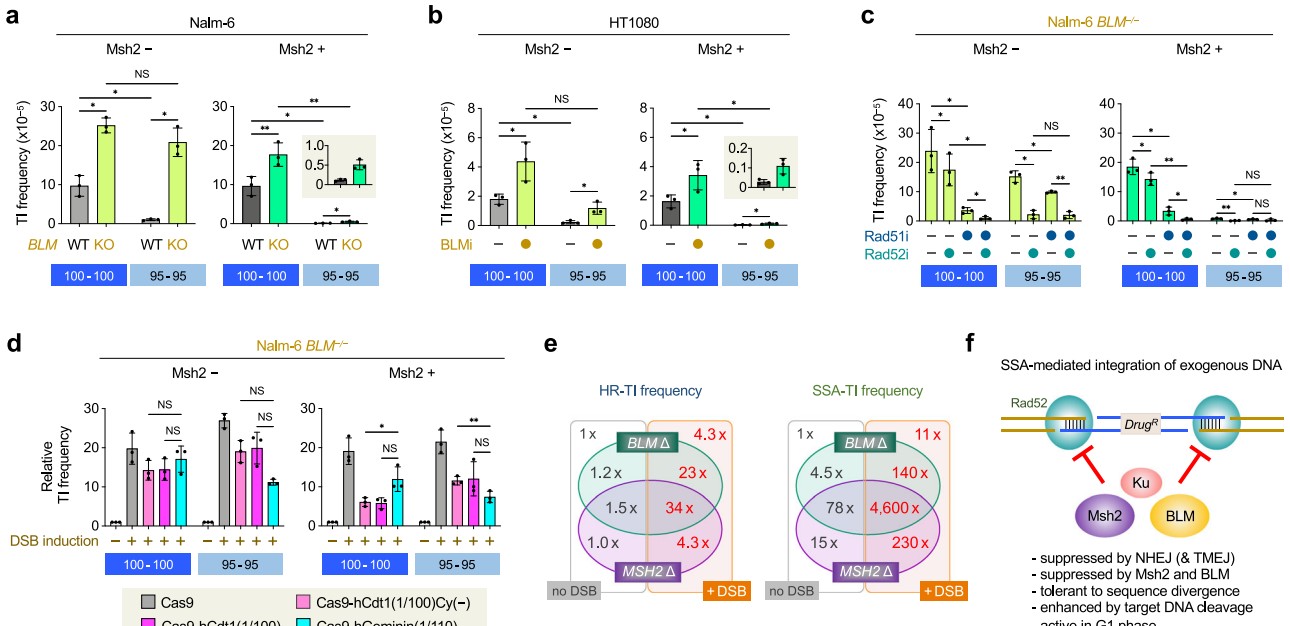

**Fig. 4 | BLM deficiency enhances SSA-mediated gene targeting. a** Impact of *BLM* gene-knockout on TI frequency. Nalm-6 Msh2- and Msh2+ cells proficient (WT) or deficient (KO) in *BLM* were transfected with p8.9HPRT-2A-Puro (100-100) or p8.9HPRT-2A-Puro-95-95 (95-95). Data shown are the mean ± s.d. (*n* = 3). **b** Impact of BLM inhibition on TI frequency. Msh2-deficient (-) and proficient (+) HT1080 cells were transfected with p8.9HPRT-2A-Bsr (100-100) or p8.9HPRT-2A-Bsr-95-95 (95-95), and treated with ML216 prior to replating. Data shown are the mean ± s.d. (*n* = 3). **c** Impact of Rad51 and Rad52 inhibition on TI frequency in *BLM⁻/⁻* cells. Cells transfected with p8.9HPRT-2A-Puro or p8.9HPRT-2A-Puro-95-95 were treated with either or both of DOPA and B02 for prior to replating. Symbols are as in Fig. 1c. Data shown are the mean ± s.d. (*n* = 3). **d** Impact of targeted DSB induction on TI frequency in *BLM⁻/⁻* cells deficient or proficient in Msh2. Cells were transfected with p8.9HPRT-2A-Puro or p8.9HPRT-2A-Puro-95-95 with or without Cas9 expression vector. The TI frequency of p8.9HPRT-2A-Puro or p8.9HPRT-2A-Puro-95-95 without Cas9 in each cell line was taken as 1, and the relative TI frequency was calculated. Data shown are the mean ± s.d. (*n* = 3). Statistical significance in (**a**–**d**) was determined by two-sided Student's t-test. *P < 0.05; **P < 0.01; NS, not significant. Source data are provided as a Source Data file. **e** Venn diagrams depicting how Msh2 loss, BLM loss, and targeted DSB induction affect gene targeting. Left, fold increase in HR-TI frequency of the 8.9-kb 100-100 vector. Right, fold increase in SSA-TI frequency of the 8.9-kb 95-95 vector. In each panel, the values were calculated from data summarized in Supplementary Fig. 13b. **f** Diagram and characteristics of SSA-TI.

the long-standing notion that gene targeting occurs via HR-TI during S/G2 phases. Intriguingly, in *LIG4⁻/⁻POLQ⁻/⁻* cells where SSA is elevated, G1-Cas9 expression resulted in an increased TI frequency of the 95-95 vector, and this increase was alleviated by the presence of Msh2 (Fig. 3g and Supplementary Fig. 9i). Intriguingly, unlike wild-type cells, *RAD54⁻/⁻RAD54B⁻/⁻* cells exhibited higher TI frequency upon DSB induction in G1 than in S/G2, even with the 100-100 vector (Supplementary Fig. 9h). These results raise the possibility that SSA-TI occurs in G1 phase more frequently than in S-G2 phases. To directly confirm this, we next used the 95-95 vector. Unlike the case of the 100-100 vector, G1-Cas9 expression resulted in a greater increase in TI frequency of the 95-95 vector than did S/G2-Cas9 expression. This was seen in wild-type and *LIG4⁻/⁻POLQ⁻/⁻* cells and, more pronouncedly, in *RAD54⁻/⁻RAD54B⁻/⁻* cells (Fig. 3g and Supplementary Fig. 9h, i). We performed similar experiments in HT1080 cells and obtained essentially the same results (Fig. 3h): TI of the 100-100 vector was enhanced by S/G2-Cas9, whereas TI of the 95-95 vector was enhanced by G1-Cas9, especially when Msh2 is absent. Taken together, we conclude that SSA-TI can occur in G1 phase of the cell cycle.

It is interesting to note that in *LIG4⁻/⁻POLQ⁻/⁻* cells, not only SSA-TI but also RI (i.e., SSA-RI) was markedly (>10-fold) enhanced by G1-Cas9 expression, and this enhancement was completely abolished by Msh2 (Supplementary Fig. 9i). Additionally, similar to the cases of SSA-TI and SSA-RI, G1-Cas9 conferred a greater effect in enhancing Alu recombination frequency than did S/G2-Cas9, and the presence of Msh2 completely abolished Alu recombination (Supplementary Fig. 7c–e). These results suggest that SSA-mediated DSB repair occurs in a cell cycle-independent manner and can cause RI as well as TI of foreign DNA in G1 phase of the cell cycle.

## BLM deficiency enhances SSA-TI as well as HR-TI

Finally, we investigated the effect of *BLM* deficiency on SSA-TI and HR-TI. Previous studies have established that loss of BLM enhances gene targeting[31,44]; however, the precise mechanism underlying this enhancement remains unclear. We performed *HPRT* gene-targeting assay using *BLM⁻/⁻* Nalm-6 cells[45] to examine the TI frequency of the 8.9-kb 100-100 and 95-95 vectors (Fig. 2d). As shown in Fig. 4a, the 100-100 vector showed a 3-fold higher TI frequency in *BLM⁻/⁻* cells than in wild-type cells. Surprisingly, the 95-95 vector showed a 20-fold higher TI frequency in *BLM⁻/⁻* cells, and this enhancement was suppressed by Msh2 expression (Fig. 4a and Supplementary Fig. 10a). Similar results were obtained using the BLM helicase inhibitor ML216[46] in HT1080 cells as well as in Nalm-6 cells (Fig. 4b and Supplementary Fig. 10b). These findings indicate that BLM loss enhances TI of the 95-95 vector more strongly than that of the 100-100 vector. Interestingly, the enhancement of 100-100 vector TI was suppressed not only by Rad51 inhibition but also Rad52 inhibition, suggesting that BLM loss enhances SSA-TI even when a 100% homologous vector is used (cf. Figs. 2e and 4c). Even more intriguingly, BLM loss-induced enhancement of 95-95 vector TI was also suppressed by Rad51 inhibition as well as Rad52 inhibition (cf. Figs. 2e and 4c). These findings suggest that, in the absence of BLM, a Rad51-dependent HR mechanism can operate to cause TI even when 5% sequence divergence is present in the targeting vector. Since mouse BLM and yeast Sgs1 are shown to function in heteroduplex rejection[26,47], it would be reasonable to speculate that human BLM exerts anti-recombinogenic activity during HR as well as SSA by suppressing heteroduplex formation between homologous sequences. This view is supported by our data showing that HR-TI and SSA-TI are similarly enhanced by BLM loss, as judged by 95-95 vector TI

 

frequency upon Rad52 or Rad51 inhibition (Supplementary Fig. 10c). A similar, albeit weak, tendency associated with BLM loss was seen with the 100-100 vector, although we observed preferential SSA enhancement by BLM loss in the GFP reporter assays (i.e., HR and SSA frequencies were 1.3- and 3.3-fold increased, respectively) (Supplementary Fig. 10d).

Additional experiments using cell cycle-dependent Cas9 expression further support the view that BLM loss stimulates TI events in a complicated manner; specifically, when BLM is absent, G1-Cas9 enhances TI of the 100-100 vector (as well as the 95-95 vector), while S/G2-Cas9 enhances TI of the 95-95 vector (as well as the 100-100 vector), and these enhancements were both suppressed by Msh2 expression and by either Rad51 or Rad52 inhibition (Fig. 4d and Supplementary Fig. 10e, f). These results suggest that DSB induction at the target locus promotes both SSA-TI and HR-TI in *BLM*[−/−] cells, regardless of cell cycle phase or sequence homology between the vector and the genome. Cell cycle-independent enhancement of HR and SSA in the absence of BLM was also observed in the GFP reporter assays (Supplementary Fig. 10g).

Based on the above results, we summarize the impact of BLM loss on the two different gene-targeting mechanisms HR-TI and SSA-TI, together with the impact of Msh2 loss and targeted DSB induction, and all combinations of these three stimulators (Fig. 4e). As mentioned above, BLM loss prominently enhances SSA-TI of the 95-95 vector (~4.5-fold when Msh2 is present and ~78-fold when Msh2 is absent) compared to HR-TI of the 100-100 vector (at most 1.5-fold regardless of Msh2 status). Targeted DSB induction confers similar outcomes to BLM loss, and the combination of these leads to an ~140-fold increased SSA-TI frequency, and a further enhancement (as high as ~4600-fold) is observed when Msh2 is also absent.

### Gene targeting via SSA-TI is feasible in HR-deficient cancer cells

To further verify the generality and utility of SSA-TI-based human cell gene targeting, we constructed 100-100 and 95-95 targeting vectors for exon 6 of the *HPRT* gene (Supplementary Fig. 11a and Supplementary Methods) and performed gene-targeting assay in Nalm-6 and HT1080 cells. The results obtained using these vectors were essentially the same as those described above, with SSA-TI being Rad51-independent, Rad52-dependent, enhanced by DSB induction in a cell cycle-independent manner, and enhanced by BLM inhibition (Supplementary Fig. 11b–d). Even more intriguingly, we additionally employed an HR-deficient human breast cancer cell line, MDA-MB-436[48], and demonstrated that precision gene targeting is indeed feasible in this slow-growing cell line (whose doubling-time is ~45 h), with all of the above characteristics of SSA-TI completely applicable (Supplementary Fig. 12a–c).

## Discussion

Homology-driven TI of exogenous DNA into the genome, generally referred to as gene targeting or targeted gene replacement, provides a powerful method for precise gene editing or genome modification. The field has long accepted that Rad51-dependent canonical HR is the sole mechanism for gene targeting in mammalian cells. Rad51 is absolutely required for HR, but does not participate in other DSB repair mechanisms including SSA; rather, absence of Rad51 results in an enhanced SSA frequency[20]. The results presented here demonstrate a noncanonical mechanism for gene targeting that relies on SSA (Fig. 4f), a highly mutagenic pathway for repairing chromosomal DNA breaks[20]. This finding is based on the fact that inhibition of Rad52, a central player in SSA, greatly reduces TI frequency in HR-compromised cells. Rad52 is a DNA binding protein that mediates homologous pairing of single-stranded DNA substrates[11,49]. Although recent studies suggest a role for Rad52 in transcription-coupled HR in mammalian cells[50,51], Rad52 is generally not required for HR reaction. In this study, we showed that Rad52 inhibition reduces SSA-TI frequency; however,

Rad52 inhibitor-treated *RAD54*[−/−]*RAD54B*[−/−] or *LIG4*[−/−]*POLQ*[−/−] cells still retained gene-targeting capacity, which was Rad51 independent. These residual rare TI events might be caused by other as yet unidentified factors and could be enhanced by BLM deficiency.

Mechanistically, it is reasonable to assume that a genomic DSB at or near the target region is a prerequisite for SSA-TI to occur, as depicted in Fig. 4f and Supplementary Fig. 13. Consistent with this assumption, SSA-TI was dramatically enhanced when a DSB was introduced at the target site by forced expression of Cas9 (Fig. 3b,c). Additionally, in cells doubly deficient in LIG4 and POLQ, HR-TI and SSA-TI occurred roughly equally without targeted DSB induction (Fig. 2f). Because the *LIG4*[−/−]*POLQ*[−/−] mutant is completely EJ deficient[3], unrepaired DSBs may provide increased opportunity for SSA to operate.

Our results revealed that SSA-TI is normally strongly suppressed by EJ pathways and BLM, and is hence a phenomenon that occurs at a low frequency (<10[−7]). In the absence of these negative regulators, however, SSA-TI frequency is dramatically enhanced, to an extent comparable to HR-TI frequency. Msh2, a key mismatch repair factor, does not affect HR-TI, consistent with an earlier study performed in human cells[28]. In stark contrast, Msh2 acts to suppress SSA-TI especially when vector arms harbor sequence divergence from the genome sequence. Previous studies have established that targeted DSB induction and cellular BLM dysfunction act to enhance gene targeting[8,31,44], and this has naturally been attributed to HR stimulation. The present results clearly reveal that both targeted DSB induction and BLM deficiency stimulate SSA-TI as well. Since BLM is a multifunctional DNA repair protein[52,53], the exact mechanism underlying SSA-TI enhancement by BLM loss awaits further studies. However, we speculate that BLM acts to negatively regulate SSA in two different ways. Firstly, BLM is very recently suggested to play a role in heteroduplex rejection during SSA repair in a Msh2-independent manner[47]; hence, it is reasonable to assume that absence of BLM facilitates the SSA reaction to proceed. Secondly, since BLM plays an important role in promoting extensive end resection[52,53], which occurs after initial (short-range) end resection during DSB repair[19,36], absence of BLM could accumulate short-range resected DSBs (see Fig. 4f and Supplementary Fig. 13). Such DSBs may not be fully competent at initiating HR and thus advantageous for SSA reaction[54], which we speculate may not require as long 3'-ssDNA as does HR. Indeed, this idea is also evidenced by our chromosomal DSB repair analysis that showed that SSA-mediated Alu recombination preferentially uses Alu elements present nearest to the cleavage site[3]. Alternatively or additionally, because it is reported that Exo1/Dna2-mediated end resection can occur when BLM is absent, DSB ends thus made could be advantageous for SSA and, less pronouncedly, HR (note that BLM loss stimulates both HR-TI and SSA-TI, as shown in Fig. 4c and Supplementary Fig. 10d).

Another important mechanistic aspect of SSA-TI is the tolerance to sequence divergence between the vector and the genome. Unlike HR-TI, whose frequency is markedly dropped by a 1% divergence present in one arm, SSA-TI permits at least 5% sequence divergence (for example, TI frequency of 95%-homology vectors in *RAD54*[−/−]*RAD54B*[−/−] cells was comparable to that of 100%-homology vectors, particularly when Msh2 is absent) (Fig. 2c and Supplementary Fig. 5). This fundamental property of SSA-TI may offer a technical advantage in human genome editing where the use of "isogenic" targeting vector is highly recommended[55,56] unless the cell line being used is mismatch repair-deficient[57]. Construction of an isogenic donor targeting vector, which is occasionally a tedious and time-consuming process, is not needed in SSA-TI-based human cell gene targeting.

Finally, and most intriguingly, the present study has established that in contrast to HR-TI that is S/G2 phase-restricted, SSA-TI is characterized by its occurrence throughout the cell cycle. SSA-TI is markedly enhanced by DSBs at the target locus, particularly when induced in the G1 phase (Fig. 3g,h). This was somewhat unexpected, but the occurrence of G1 (or

G0) end resection has been reported by several groups[37,38]. Additionally, homology-mediated gene targeting was reported in mature postmitotic neurons[58], although that study awaits mechanistic insights into possible TI events observed in the postmitotic cells. It is also worth mentioning that earlier studies in yeast showed a competition between Rad52 and Ku at DSB ends, implying a role for Rad52 in G1[39]. Indeed, we revealed that chromosomal DSB repair by SSA was somewhat more active in G1 than in S/G2 (Fig. 3f and Supplementary Fig. 7c, d). The finding that SSA-TI can occur before S phase, particularly when the target site has a DSB, implies that precision gene targeting is feasible in cells with a limited number of divisions such as normal somatic cells, in cells that are deficient in HR (e.g., BRCA1/2-defective cancers, as demonstrated in this study), and possibly in cells that have lost their mitotic potential due to differentiation (e.g., neurons or cardiac muscle cells). Although SSA-TI is suppressed by multiple mechanisms, targeted DSB induction as well as transient suppression of Msh2 and/or BLM will facilitate SSA-based precise gene editing.

In summary, we have revealed the existence of a noncanonical gene targeting mechanism that is active in G1 phase of the cell cycle and does not require isogenic DNA constructs. Our findings will contribute to a better understanding of foreign DNA integration and the further development of efficient genome editing technology in mammalian cells.

## Methods

### Cell culture and transfection

All cell lines used in this study were cultured in a humidified atmosphere of 5% $CO_2$ incubator at 37 °C, and tested for mycoplasma contamination using a Venor®GeM OneStep Mycoplasma Detection Kit (Minerva Biolabs Inc., Berlin, Germany). The human pre-B cell line Nalm-6 (S14)[28,45] and its derivatives were cultured in Eagle's Minimum Essential Medium (MEM; Nissui Seiyaku, Tokyo, Japan) supplemented with 10% heat-inactivated calf serum (Cytiva, Tokyo, Japan), MEM Non-Essential Amino Acids Solution (FUJIFILM Wako Pure Chemical, Osaka, Japan), 1 mM sodium pyruvate, 50 μM 2-mercaptoethanol (FUJIFILM Wako Pure Chemical), and 0.15 μM vitamin B12 (Sigma-Aldrich, St. Louis, MO, USA)[3]. The human fibrosarcoma cell line HT1080 was obtained from Institution for Fermentation (Osaka, Japan), and cultured in Dulbecco's Modified Eagle Medium (Nissui Seiyaku) supplemented with 10% heat-inactivated calf serum[31]. Nalm-6 and HT1080 were authenticated by short tandem repeat analysis (Promega, Fitchburg, WI, USA). The human breast adenocarcinoma cell line MDA-MB-436 was obtained from American Type Culture Collection (HTB-130, ATCC, Manassas, VA, USA), and cultured in RPMI-1640 medium (FUJIFILM Wako Pure Chemical) supplemented with 10% heat-inactivated calf serum.

DNA transfection using the MaxCyte STX device (MaxCyte, Rockville, MD, USA) was performed according to the manufacturer's instructions. Briefly, $1 \times 10^7$ cells were suspended with 100 μl of the supplied solution (MaxCyte Electroporation Buffer) and transfected with 5 or 10 μg of DNA[3]. DNA transfection using the GTE-1 electroporation apparatus (Shimadzu, Kyoto, Japan) was performed according to the manufacturer's instructions. Briefly, cells were washed twice with Saline G (130 mM NaCl, 5.3 mM KCl, 1.1 mM $Na_2HPO_4$, 1.1 mM $KH_2PO_4$, 6.1 mM glucose, 0.49 mM $MgCl_2$ and 0.9 mM $CaCl_2$) and an aliquot of the cell suspension ($4 \times 10^6$ cells in 40 μl of Saline G) was electroporated with 4 μg of DNA[3]. DNA transfection using the Nucleofector II system (Lonza, Basel, Switzerland) was performed according to the manufacturer's instructions. Briefly, $6 \times 10^6$ cells were suspended in 100 μl of the supplied solution (Solution R), and transfected with 12 μg of DNA.

### Nalm-6 gene-knockout mutant cell lines

All the gene-knockout mutant cell lines from Nalm-6 were generated by conventional gene targeting without using CRISPR/Cas9.

$RAD54^{-/-}RAD54B^{-/-15}$, $LIG4^{-/-59}$, $POLQ^{-/-3}$, $LIG4^{-/-}POLQ^{-/-3}$, and $BLM^{-/-45}$ cells have been previously described, and the knockout status of these cells was verified by western blot analysis before use (see Supplementary Fig. 3b).

Targeting vectors for the human *RAD52* gene were constructed as described previously[3]. Briefly, 1.8 and 4.6-kb *RAD52* genomic fragments were PCR amplified with Tks Gflex DNA Polymerase (Takara Bio, Otsu, Japan) using Nalm-6 genomic DNA as a template. The primers used were Rad52-5-Fw and Rad52-5-Rv for the 1.8-kb 5′ arm, Rad52-3-Fw and Rad52-3-Rv for the 4.6-kb 3′ arm. By using In-Fusion®HD Cloning Kit (Clontech, CA, USA), a *PGK* promoter-linked puromycin-resistance gene (*Puro^R*) or hygromycin-resistance gene (*Hyg^R*) was placed between the 5′ and 3′ arms, thus yielding targeting vectors pRad52-Puro and pRad52-Hyg. To disrupt the *RAD52* gene, linearized pRad52-Hyg was transfected into Nalm-6 cells and hygromycin-resistant colonies were isolated and expanded to prepare genomic DNA. Gene-targeting events were screened by PCR analysis using primers Rad52-5′ ext and Universal primer Hyg. Subsequently, linearized pRad52-Puro was transfected into *RAD52^{+/-}* cells, and puromycin-resistant clones were subjected to PCR analysis using primers Rad52-5′ ext and Rad52-Nega. PCR primers used to generate Rad52 knockout cells are listed in Supplementary Table 1. The disruption of the *RAD52* gene was further confirmed by western blot analysis using anti-Rad52 antibody.

### Creation of *MSH2*-corrected Nalm-6 cell lines

Restoration of Msh2 expression in Nalm-6 cells was performed essentially as described[28]. Briefly, PmeI-linearized pMSH2-Neo vector was transfected into wild-type, *RAD54^{-/-}RAD54B^{-/-}*, *LIG4^{-/-}POLQ^{-/-}*, *RAD52^{-/-}*, and *BLM^{-/-}* cells using the GTE-1 electroporation apparatus, and G418-resistant colonies were isolated and expanded to prepare genomic DNA. Correct gene-targeting events were screened by PCR analysis using primers MSH2 GT-Fw and 3′-loxP (Supplementary Table 1). The restoration of Msh2/Msh6 expression was confirmed by western blot analysis.

### Generation of *MSH2*-knockout HT1080 cells

pX330-Cas9-MSH2 (Supplementary Table 2) and AhdI-linearized pPGKpuro[60] were co-transfected into HT1080 cells using the Max-Cyte STX device, and puromycin-resistant colonies were isolated and expanded to prepare genomic DNA. The disruption of the *MSH2* gene was confirmed by PCR analysis using primers MSH2-Del-Check-Fw and MSH2-Del-Check-Rv (Supplementary Table 1). The PCR products were cloned into the pTAKN2 T-Vector (BioDynamics Laboratory Inc., Tokyo, Japan) to determine the sequence (Eurofins Genomics K.K., Tokyo, Japan). The disruption of the *MSH2* gene was further confirmed by western blot analysis using anti-Msh2 and anti-Msh6 antibodies.

### *HPRT* targeting vectors

*HPRT* targeting vectors used in this study (listed in Supplementary Table 3) were designed to insert a 2A peptide-linked *Puro^R* or blasticidin-resistance gene (*Bsr*) cassette into exon 3 or exon 6 of the human *HPRT* gene. Genomic DNA fragments for the arms were PCR amplified with Tks Gflex DNA Polymerase using Nalm-6, HT1080, or MDA-MB-436 genomic DNA as a template. (Note that isogenic DNA constructs were employed in to avoid the effect of sequence divergence (SNPs) on gene targeting[55,56].) Vector construction was performed using the MultiSite Gateway system (Life Technologies, Rockville, MD, USA)[59,61,62] or In-Fusion cloning[3], placing a 2A peptide-linked *Puro^R* or *Bsr* between the 5′ and 3′ arms. p8.9HPRT-2A-Bsr was constructed using In-Fusion®HD Cloning Kit. Briefly, a 1.0-kb fragment containing 2 A peptide sequence, a blasticidin-resistance gene (*Bsr*), and polyA sequence was PCR amplified with PrimeSTAR MAX DNA Polymerase (Takara Bio) using pUC-Bsr (obtained as synthetic DNA fragments (GenScript Japan K.K., Tokyo, Japan)) as a template and primers HPRT-2A-Bsr-Fw and HPRT-2A-Bsr-Rv (Supplementary

Table 1). The PCR fragment and a XhoI-digested p8.9HPRT-2A-Puro[15] were subjected to In-Fusion cloning, yielding p8.9HPRT-2A-Bsr.

Targeting vectors harboring mismatches (base substitutions) in the arms were generated by standard molecular biology techniques and artificial gene synthesis (GenScript Japan K.K., Tokyo, Japan). PCR primers used to construct *HPRT* targeting vectors are listed in Supplementary Table 4, and details of construction and sequence information for mutated homology arms are described in Supplementary Methods.

Targeting vectors with 20-bp arms (HPRT-2A-Puro-40bp), 40-bp arms (HPRT-2A-Puro-80bp), and 103-bp and 109-bp arms (HPRT-2A-Puro-200bp) were PCR amplified with Tks Gflex DNA Polymerase using p3.0HPRT-2A-Puro as a template. The primers used were HPRT-20bp-Fw and HPRT-20bp-Rv for HPRT-2A-Puro-40bp, HPRT-40bp-Fw and HPRT-40bp-Rv for HPRT-2A-Puro-80bp, and HPRT-100bp-Fw and HPRT-100bp-Rv for HPRT-2A-Puro-200bp (Supplementary Table 1).

All the plasmid targeting vectors were purified with Qiagen Plasmid *plus* Midi Kits (Qiagen K.K., Tokyo, Japan) and linearized with an appropriate restriction enzyme prior to transfection[61]. All the PCR-based targeting vectors were purified with Wizard® SV Gel and PCR Clean-Up System (Promega) prior to transfection.

## Chemicals
Puromycin, hygromycin B, G418, and blasticidin S were purchased from FUJIFILM Wako Pure Chemical and dissolved in distilled water. B02, RI-2, ML216, and 6-thioguanine (6TG) were purchased from Sigma-Aldrich, and 6TG was dissolved in 0.5% sodium carbonate and the others in dimethyl sulfoxide. 6-Hydroxy-DL-DOPA (DOPA) was purchased from R&D Systems (Minneapolis, MN, USA) and dissolved in 10% hydrochloric acid. AICAR was purchased from Merck Millipore (Burlington, MA, USA) and dissolved in dimethyl sulfoxide.

## shRNA expression vectors
A DNA fragment containing an shRNA sequence for Rad52 (5′-GGAAATATGATCCATCTTA-3′) or a negative control shRNA (5′-CCTAAGGTTAAGTCGCCCT-3′) was prepared by annealing oligonucleotides shRad52-top and shRad52-btm, or shCtrl-top and shCtrl-btm, respectively. Each DNA fragment was cloned into pBAsi-hU6 Neo (Takara Bio). Oligonucleotides used to construct shRNA expression vectors are listed in Supplementary Table 1.

## I-SceI expression vectors
Cell cycle-regulated I-SceI expression vectors were generated by standard molecular biology techniques. Briefly, a 1.2-kb BsrGI/DraI fragment containing human Geminin encoding amino acid residues 1-110 (hGeminin(1/110)) from pSce-Cy-G2[63] or a 1.1-kb BsrGI/DraI fragment containing human Cdt1 encoding amino acid residues 30-120 (hCdt1(30/120)) from pSce-Cy-G1[63] was subcloned into a BsrGI/MscI-digested pSceI plasmid[45], thus yielding pSceI-Geminin and pSceI-Cdt1, respectively.

## Cas9 expression vectors
Cas9 expression vectors used in this study are summarized in Supplementary Table 2. Cell cycle-regulated Cas9 expression vectors (pX330-Cas9-hGeminin(1/110), pX330-Cas9-hCdt1(1/100), pX330-Cas9-hCdt1(1/100)Cy(-), and pX330-Cas9-hCdt1(30/120)) were constructed by fusing a partial cDNA of either human Geminin encoding amino acid residues 1-110 (hGeminin(1/110)), human Cdt1 encoding amino acid residues 1-100 (hCdt1(1/100)), human Cdt1 encoding amino acid residues 1-100 with mutations (R68A, R69A, L70A) in the cyclin-binding motif (hCdt1(1/100)Cy(-)), or human Cdt1 encoding amino acid residues 30-120 (hCdt1(30/120)) to Cas9 cDNA[40,64]. Oligonucleotides used to construct Cas9 expression vectors are summarized in Supplementary Table 1, and details of construction are described in Supplementary Methods.

## Gene-targeting assay
Gene-targeting assay was performed as described previously[3,62,65]. For Nalm-6 assays, transfected cells were cultured for 24 h and plated into agarose medium. After a 2–3 week incubation, the resulting colonies were counted, and the total integration frequency was calculated by dividing the number of drug-resistant colonies by the number of surviving cells. Subsequently, single colonies were isolated, expanded, and replated in growth medium containing 20 μM 6TG, a hypoxanthine analog that kills HPRT-proficient cells. Genomic DNA was isolated from 6TG-resistant clones and subjected to PCR analysis. Gene-targeting efficiency was calculated by dividing the number of targeted clones by the number of puromycin-resistant clones analyzed. TI frequency was calculated by multiplying integration frequency by gene-targeting efficiency. RI frequency was calculated by subtracting TI frequency from integration frequency. For HT1080 assays, transfected cells were cultured for 48 h and replated at a density of $2-5 \times 10^5$ cells per 90-mm dish. After a 24-h incubation, blasticidin S was added to the 90-mm dishes (10 μg/ml), and cells were selected with 20 μM 6TG. After 7 days of incubation, genomic DNA was isolated from blasticidin-resistant colonies and subjected to PCR analysis. The frequency of TI was calculated by dividing the number of targeted clones by that of surviving cells. For MDA-MB-436 assays, transfected cells were cultured for 48 hr and replated at a density of $5-10 \times 10^5$ cells per 90-mm dish. After a 24-h incubation, puromycin was added to the 90-mm dishes (0.4 μg/ml), and then cells were selected with 20 μM 6TG. After 3 weeks of incubation, genomic DNA was isolated from puromycin-resistant colonies and subjected to PCR analysis. The frequency of TI was calculated by dividing the number of targeted clones by the number of surviving cells.

For experiments using chemical inhibitors, transfected cells were transferred into growth medium containing Rad51 inhibitor (10 μM B02 or 75 μM RI-2), Rad52 inhibitor (10 μM DOPA or 10 μM AICAR), or BLM inhibitor (5 μM ML216) and cultured for 24 hr. Cells were then washed with PBS⁻ and cultured for 1–3 weeks to allow for colony formation. For shRNA-mediated gene-knockdown experiments, cells were transfected with either pshRad52 or pshControl, and cultured for 48 hr in growth medium containing 1.0 mg/ml G418. For siRNA-mediated gene-knockdown experiments, cells were transfected with either siXPF (L-019946-00; Horizon Discovery, Cambridge, UK), siERCC1 (L-006311-00; Horizon Discovery), or siControl (D-001810-10; Horizon Discovery), and cultured for 24 h. The cells were then electroporated with a targeting vector, cultured for 24 h and plated into agarose medium. For determination of knockdown efficiencies, aliquots of the cell suspension were subjected to western blot analysis. For gene-targeting assay using the CRISPR/Cas9 system, cells were co-transfected with equal amounts of targeting vector and either pX330-Cas9-HPRT-Ex3, pX330-Cas9-hGeminin(1/110)-HPRT-Ex3, pX330-Cas9-hCdt1(30/120)-HPRT-Ex3, pX330-Cas9-hCdt1(1/100)-HPRT-Ex3, pX330-Cas9-hCdt1(1/100)Cy(-)-HPRT-Ex3, pX330-Cas9-HPRT-Ex6, pX330-Cas9-hGeminin(1/110)-HPRT-Ex6, pX330-Cas9-hCdt1(1/100)-HPRT-Ex6, or pX330-U6-Chimeric_BB-CBh-hSpCas9 (Supplementary Table 2).

## GFP reporter assay
DR-GFP/SA-GFP reporter assays were performed as previously described[66,67]. Briefly, cells with the GFP reporter cassette (see Supplementary Methods for details) were transfected with 2 μg of pSceI, pSceI-Geminin, pSceI-Cdt1, or pmaxGFP (Lonza (Basel, Switzerland)) using the MaxCyte STX device. For the DR-GFP reporter assay using the CRISPR/Cas9 system, pX330-Cas9-DR, pX330-Cas9-hGeminin(1/110)-DR, pX330-Cas9-hCdt1(30/120)-DR, pX330-Cas9-hCdt1(1/100)-DR, or pX330-Cas9-hCdt1(1/100)Cy(-)-DR was used to induce a DSB, and for the SA-GFP reporter assay using the CRISPR/Cas9 system, pX330-Cas9-SA, pX330-Cas9-hGeminin(1/110)-SA, pX330-Cas9-hCdt1(30/120)-SA, pX330-Cas9-hCdt1(1/100)-SA, or pX330-Cas9-hCdt1(1/100)Cy(-)-SA

was used (see Supplementary Table 2). Cells were then cultured for 72 h, and GFP-positive cells were counted using an Attune Nxt Flow Cytometer (Thermo Fisher Scientific, Waltham, MA, USA). For experiments using chemical inhibitors, cells were cultured for 24 h in the inhibitor-containing growth medium, washed with PBS⁻, and cultured for an additional 48 h. In each experiment, $1–5 \times 10^5$ cells were analyzed, and the percentage of GFP-positive cells was calculated from the number of GFP-positive cells divided by the number of cells analyzed.

### Western blot analysis

Cells ($2–10 \times 10^6$) were washed twice with PBS⁻ and scraped in 150 μl of lysis buffer (50 mM Tris-HCl (pH 6.8), 2% sodium dodecyl sulfate, 10% glycerol, 100 μM dithiothreitol, 1 mM phenylmethylsulfonyl fluoride) containing Protease Inhibitor Cocktail (Sigma-Aldrich). The lysates were allowed to stand for 20 min at 4 °C and, after sonication, centrifuged at $21,500 \times g$ for 30 min at 4 °C. The supernatants were collected and used for western blot analysis. Protein concentration was determined using the Protein Assay BCA Kit (FUJIFILM Wako Pure Chemical). Fifteen micrograms of the lysates were electrophoresed in a 7.5% polyacrylamide gel or a 5–15% gradient polyacrylamide gel (Funakoshi, Tokyo, Japan) and then transferred onto a polyvinylidene difluoride membrane (Merck Millipore). Membranes were blocked with 5% skim milk and then incubated with a primary antibody, followed by incubation with a horseradish peroxidase-conjugated secondary antibody. Signals were detected with Clarity Western ECL Substrate (Bio-Rad, Hercules, CA, USA) and analyzed using a Fuji Image Analyzer LAS-1000UVmini (FUJIFILM Co., Tokyo, Japan). The antibodies used are listed in Supplementary Table 5.

Western blot analysis of sorted cells was performed essentially as described[68]. Briefly, HT1080 cells were transfected with either pX330-Cas9-HPRT-Ex3, pX330-Cas9-hCdt1(1/100)-HPRT-Ex3, or pX330-Cas9-hGeminin(1/110)-HPRT-Ex3. After a 48-h incubation, cells were incubated for 1 h in growth medium containing 5 μg/ml Hoechst 33342 (DOJINDO Laboratories, Kumamoto, Japan), detached with trypsin, and resuspended in growth medium containing 5 μg/ml Hoechst 33342. Stained cells were sorted according to their DNA content using Cell Sorter SH800S (Sony Corporation, Tokyo, Japan). Approximately $10^6$ cells were collected per G1 and S/G2/M phase. After sorting, cells were washed once with growth medium, and then subjected to western blot analysis.

### Statistics and reproducibility

Data analysis was performed using GraphPad Prism (version 8.4.3) and Microsoft Excel (version 16.16.11). A $P$ value less than 0.05 was considered significant. All experiments were independently replicated, as described in each figure legend, and no data was excluded from the study.

### Reporting summary

Further information on research design is available in the Nature Portfolio Reporting Summary linked to this article.

## Data availability

All data that support the findings of this study are available within the article and its Supplementary Information. Source data are provided with this paper.

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

## Acknowledgements

We thank Takehiko Nohmi and Tetsuya Suzuki for providing us with the pMSH2-Neo plasmid. We also thank Takaaki Yasuhara and Kiyoshi Miyagawa for generously providing us with anti-Rad54B antibody. We are deeply thankful to Andrew Porter, who sadly passed away in August 2023, for I-SceI expression vectors. We would like to dedicate this paper to Andy. We are very grateful to Kaoru Okamura and Yoshiko Hosokawa for their excellent technical assistance. We also thank Usaki Arai, Naoko Fujimoto, Shota Morimoto, and Shigeki Sueyoshi for their technical assistance. This work was supported by JSPS KAKENHI Grant Numbers JP18K19407 (N.A.), JP19H01151 (N.A.), JP22K19382 (N.A.) and JP21K15069 (S.S.), and by Yokohama City University Strategic Research Promotion Grant Number SK201901 (N.A.).

## Author contributions

S.S. performed experiments and analyzed the data. N.A. designed the study and analyzed the data. N.A. and S.S. wrote the manuscript.

## Competing interests

The authors declare no competing interests.
