## [Peer Review File · Nature Communications]

Characterization and regulation of cell cycle independent noncanonical gene targetingREVIEWER COMMENTS

Reviewer #1 (Remarks to the Author):

Saito and Adachi show that targeted integration in mammalian cells, whether or not induced by DSB-generation using Cas9, can take place via a Rad52-mediated mechanism. This SSA-TI, as the authors call it, also takes place during the G1-phase of the cell-cycle and can still occur even if sequence homology between target site and template is not 100%.

I appreciate that the authors make an extensive effort to investigate the nature of Rad51-independent DNA integration. However, the finding that this is through a Rad52-dependent mechanism might not be very surprising, given the fact that this is the other homology-dependent recombination pathway present in mammalian cells. More importantly, some of the main conclusions, in particular concerning SSA-TI primarily taking place in G1, are not sufficiently supported by the data, as I will explain in more detail below.

Major comments:

1. One of the main conclusions is that a fraction of TI is Rad52-dependent. To support this, the authors show that treatment with two Rad52 inhibitors blocks TI in HR-deficient cells. However, both AICAR and DOPA are rather unspecific drugs that for example activate AMPK (AICAR) and inhibit APE1 (DOPA). The authors validate the involvement of Rad52 with a Rad52 shRNA, but the knockdown observed with this shRNA is very poor (extended data 2a). To show conclusive evidence for this main conclusion, the authors should show a TI phenotype in cells that lost Rad52 expression using either a properly working shRNA or CRISPR-generated Rad52 KO cells.
2. Furthermore, whereas the authors are at times careful to suggest that the Rad52-dependent TI is SSA, they do strongly suggest this at other times, and also refer to it as SSA-TI. To more firmly conclude that the Rad51-independent integration is in fact SSA, the authors should assess SSA-frequency upon loss of another SSA-factor, like XPF/ERCC1. This might require including a non-homologous region in the repair template, but this can easily be engineered and would allow to test the SSA-hypothesis.
3. Based on the results presented in figure 3d-h and ext. data 8, the authors conclude that SSA is preferably active in G1 (lines 383-386). I think the data are insufficient to support this conclusion, based on the following:
 - a. In all SSA reporter assays, levels of SSA are comparable when using G1-Cas9 or S/G2-Cas9 (fig. 3f, ext. data 8b), and even higher for S/G2-I-SceI than for G1-I-SceI (ext. data 8c).
 - b. Using TI frequency with the 95-95 template as a read-out for SSA, G1-Cas9 performs better than S/G2-Cas9 in some assays (Nalm-6 WT, HT1080), but not in others (Nalm-6 LIG4/POLQ^{-/-}). The data for the Nalm-6 RAD54^{-/-} cells are n=1 and therefore inconclusive.
 - c. The system is leaky, as there might still be some expression of G1-Cas9 in S/G2 (and the other way around), and a break generated in G1 might be repaired in S/G2.
 - d. The FUCCI-Cas9 system is not validated, so expression levels of the different variants might be incomparable and not as cell-cycle specific as suggested by the inferred data shown in panel 3d.
4. To solve this, the authors should at least:
 - a. Assess the protein levels of the FUCCI-Cas9 variants throughout the different cell-cycle phases to validate their system.
 - b. Repeat the TI assays in the Nalm-6 RAD54^{-/-} cells to be able to make a statistical comparison between G1-Cas9 and S/G2-Cas9.
 - c. Explain why some of the TI assays suggest that SSA might be more dominant in G1, while the SA-GFP reporter assays do not show this phenotype.
 - d. Tone down the conclusion, as the results indicate that SSA might also occur in G1, but are not sufficient to conclude that SSA preferentially occurs in G1.
5. Although technically challenging, the authors could use inducible systems like iCas9 or UV-inducible

guide RNAs in combination with cell-cycle synchronization methods to further validate G1-specific SSA. 6. Based on fig. 4a, b, the authors conclude that BLM-loss enhances SSA-TI more so than HR-TI. However, this cannot be concluded, as the 95-95 template can also integrate by HR-TI in the BLM-/- background, as shown in fig. 4c. Thus, the stronger fold-change in TI-frequency when using the 95-95 template is caused by a combined effect on SSA-TI and HR-TI. To conclude that BLM loss affects SSA-TI more than HR-TI, the authors should compare TI frequency between WT and BLM-/- cells that are treated with Rad51i or Rad52i.

7. The authors suggest that SSA-TI could be used for gene targeting in postmitotic cells or HR-deficient cells. The authors could show this to demonstrate the applicability of their findings. I would suggest to perform Cas9-mediated gene editing in HR-deficient cells like MDA-MB-436 or PEO-1 (eg by GFP-tagging an endogenous gene), and demonstrate that this is dependent on Rad52, and can be enhanced by BLM inhibition.

Minor comments:

1. Are the PCR-products to assess TI in the puro- and 6TG-resistant colonies always exactly the predicted size? Imaginably, SSA-TI could be a bit more flexible and occur upon imperfect annealing resulting in deletions.
2. Line 151-152: "This may imply that a lesser extent of end-resection is sufficient to initiate SSA". This is an unlikely explanation, as most studies would suggest that SSA is more dependent on long range end-resection factors than HR (see for example van de Kooij et al. Nat Commun 13, 2022 and Tomimatsu et al., J Biol Chem 292, 2017).
3. Line 171-173: "TMEJ does not have any role in HR-independent TI events". In fact, MMEJ-dependent Cas9-TI has been demonstrated to occur (Nakade S et al. Nat Commun 9, 3270 (2018)). Whereas loss of POLQ does not affect TI in Nalm-6 cells, this might be dependent on the repair template, or cell-line, used. The authors should comment on this.
4. I am not sure what the value is of showing gene targeting efficiency. If I understand correctly, it looks at all integrations within the gene (because 6-TG resistant), which can be either targeted, or random. I would argue that showing the frequency of TI vs RI is sufficient, and whether or not there is also random integration that happens to occur within the gene is less relevant, and perhaps even confusing.
5. The paper would benefit from reordering the figures to prevent going back and forth too much. For example, fig. 1e is discussed before 1d, extended data 2 before most of extended data 1.

Reviewer #2 (Remarks to the Author):

In this manuscript, Saito and Adachi present a very comprehensive and detailed study into the biology underlying gene targeting. They uncover a mechanism alternative to Rad51-guided HR that is able to target ectopically provided DNA to genomic loci. This SSA mechanism (termed SSA primarily because it is Rad52 dependent) is stimulated by DNA breaks at the locus and suppressed by the repair factors Msh2 and Blm. Of particular interest is the observation (and careful experimental design) that Cas9 fusion proteins that are expressed at specific cell cycle stages affect the SSA-mediated targeting differently. It is concluded that the mechanism (can) acts in G1, which would also be of potential clinical importance given the future genome editing may require the targeting of non-dividing cells.

All in all, I find the paper impressive, both on the depth, the quality of the experimental design and data, the clarity of the writing. I also find the novelty more than sufficient to warrant publication in a journal with broad readership (as is Nat. commun). I thus fully support publication of a manuscript that incorporates the underneath.

The only "major" issue I have is whether the authors can be more convincing (perhaps I missed the

argument or the experiment in support) that SSA-mediated targeted integration at break sites actually takes place in G1 (as opposed to being initiated in G1). They demonstrate that the frequency goes up when DSBs are generated in G1 but that doesn't necessarily mean that SSA is happening in G1. I could also envisage a scenario where DSBs generated in G1 are processed such (for instance upon ensuing DNA replication when cells progress to S) that they have a different configuration compared to DSB that are generated in S/G2. I could think of very complex technically challenging experiments, which may (or may not) provide an answer to this question, but feel that this would be beyond what is reasonable to ask. However, if the authors are unable to provide evidence that SSA-TI is "finished" while cells are still in G1, I would find it more reasonable to i) adapt the text to accommodate this possibility (in abstract and throughout the text), and discuss this option in e.g. discussion section.

Minor issues:

- i) I feel the title "Compelling evidence" is rather unconventional and also a somewhat unscientific/subjective. Merely removing "Compelling" doesn't do the trick. One would hope that all papers have "evidence" for what is presented.
- ii) Please specify the Extended figure panels where appropriate. The figures are quite extensive and it would help the reader to point to the specifics: For instance in line 110, refer to Extended Data Fig. 1A+B, as opposed to only Fig. 1), and so forth.
- iii) Line 142, I could not find the 1.5- fold (in Ex Fig 1F).
- iv) Line 151-152 holds unnecessary speculation, which I don't agree with: to me, the mechanism for HR is much more complex than SSA, as it needs DNA synthesis on a template (first processing of the flap prior to DNA synthesis) and therefore there are many potential reasons for why SSA is more rapid, not at all pointing to end-resection (kinetics).
- v) Line 162: wouldn't 1.5-2 fold be more appropriate (2 fold is somewhat overly optimistic given the data presented in Ext Figs 1E and 1C).

I would like to end with complimenting the authors on this impressive piece of work (impressive also considering that just one person carried out all experiments).

Reviewer #3 (Remarks to the Author):

In this manuscript, Saito & Adachi used gene targeting to the HPRT gene, together with the DR-GFP and SA-GFP reporters, to analyze the role of SSA in homology-directed gene targeting. The question asked by the authors has been asked and addressed before, but this study presents some new findings: 1) Rad52 inhibition suppresses gene targeting and homology-directed repair (HDR) in HR-deficient cells, not in HR-proficient cells, suggesting involvement of SSA; 2) This SSA mechanism tolerates 5% divergence in homology arms but HR does not, thus giving Msh2 more suppressive control in SSA; 3) SSA-mediated targeted and random integration also occurs in G1 in addition to S/G2; 4) BLM loss has more significant impact on SSA-mediated TI than on HR-mediated TI. These findings are very interesting. In particular, G1 occurrence of SSA may help clarify the view on the cell cycle stage for SSA although the exact regulation is yet to be determined. In this regard, the evidence is convincing to me but additional experiments are needed to solidify this point as explained below. The authors also stated that this study has implications for precise gene editing in post-mitotic cells. However, the level of SSA-mediated TI is quite low, limiting practical use of this strategy in precise gene editing and by extension the significance of this manuscript. Additional comments are also detailed below.

Detailed comments:

1. The title of this manuscript is vague and does not truly reflect the key findings of this study.
2. The efficiency of SSA-mediated TI is much lower by up to 25-fold than Rad51-mediated TI. Thus, applications of SSA-mediated TI in genome editing could be very limited to non-growing cells and insignificant in proliferating cells. In addition, increasing the efficiency of SSA-mediated TI is still

needed for practical use even in non-growing cells. It would strengthen the manuscript if this study had tested SSA-mediated TI by quantitative PCR and sequencing in slow-growing cells or even non-growing cells.

3. The targeting vector contains a promote-less 2A-Puro cassette. For RI, the cells would survive with puromycin selection as long as the 2A-puro is inserted into coding regions of genes within a correct reading frame. If this happens, the level of RI could be greatly underestimated in this assay.

4. In this manuscript, it is unclear how many copies of the DR-GFP and SSA-GFP reporter stably integrated in cells. The authors need to quantify the copy number of these reporters randomly integrated and restrict copies of the DR-GFP and SSA-GFP reporter in cells in order to avoid the undesired interference of homologous copies with SSA and HR.

5. The authors used only one site for targeted integration. If this site has some particular nucleotide composition and chromatin context, which could affect the control of targeted integration, it is possible that analysis in this manuscript could not be generalized. I would suggest additional validation at one more site away from the site used already in the manuscript.

6. The authors used a few KO cell lines in this study but did not provide details how these cell lines or particular clones were generated, confirmed and chosen.

7. In Extended Data Fig. 2a, c, Rad52 shRNA used is not that effective. Is it a reason why there exists inconsistency in the effect on gene targeting efficiency between Rad52i treatment and Rad52 shRNA in Extended Data Fig. 2b, d?

8. Fig. 1e is presented again in Extended Data Fig. 2d.

9. In Line 71-74, "100% efficiency of gene targeting" is misleading and should be reworded.

10. In Line 85-87, "less than 5% of Rad54/Rad54B-proficient cells" is confusing and should be reworded as "less than 5% of the frequency in Rad54/Rad54B-proficient cells".

11. In Line 171-173, the statement "It should be noted, however, that TMEJ, unlike SSA (which requires a longer sequence of homology than TMEJ), does not have any role in HR-independent TI events" is not accurate. In the gene targeting assays, the TI frequency was slightly higher in LIG4 KO/POLQ KO cells treated with Rad51 inhibitors, but the TI levels were not determined in POLQ KO cells. Therefore, this conclusion could not be drawn.

12. In Extended Data Fig. 7a, site-specific DSB induction in the genome promotes RI at other genomic regions. Why?

13. In Extended Data Fig. 7d, as TI with short homology arms (80 bp and 212 bp) is mediated primarily by Rad52-dependend SSA and suppressed by Msh2, it is expected that 5% divergence in short homology arms would not affect TI with short homology arms (80 bp and 212 bp). But is it really the case? No data were provided.

14. The authors used the Fucci system to control the Cas9 protein level in specific cell cycle phase and intended to induce site-specific DSB at a cell cycle phase-specific manner. However, as demonstrated previously, Cas9 can remain tightly bound to cleaved target for a long period of time (up to 6-7 hours), affecting DNA repair (Stenberg et al, Nature, 2014, 506:62-67; Richardson et al, Nat Biotech, 2016, 34:339-344; Liu et al, Genome Biol, 2022, 23:165). Thus, DSBs induced by Cas9 in the G1 phase could be exposed only later in the S phase for DNA repair, interfering with the analysis in this study. In addition, although the Cas9 protein level is controlled by hCdt1 degradation in a cell cycle-specific manner, it is unclear whether target-bound Cas9 can be degraded before being released from its targets. It is likely that in order to degrade target-bound Cas9 in proteosomes, release from its targets in the context of chromatin and getting into the cytoplasm from the nucleus are necessary. In addition, it is unclear whether Cas9 cleaves DNA in similar efficiencies between different cell cycle stages. Therefore, the authors should provide evidence in this regard by analyzing both the cell cycle distribution of Cas9 protein and the efficiency of Cas9-induced DNA cutting in different cell cycle phases.

15. In Line 383, the authors stated: "Surprisingly, G1-Cas9 expression showed a greater effect in enhancing SSA than did S/G2-Cas9 expression, suggesting that SSA-mediated DSB repair operates in G1, rather than outside G1 phase (Fig. 3f and Extended Data Fig. 8b)." Given a considerable level of SSA outside the G1 phase, this statement is inaccurate.

16. In Line 410-411, the statement "Taken together, we conclude that SSA-TI primarily occurs in the G1 phase of the cell cycle" is not accurate as SSA and SSA-TI are robust and comparable in both G1

and S/G2 (Fig. 3f, g and Extended Data Fig. 8).

17. In many cases (e.g., Fig. 3g, Fig. 4d and Extended Data Fig. 2d), the difference in results between three independent experiments is too large, resulting in large standard deviation, any conclusion determined by statistical analysis with one way or another could be misleading unless no conclusion is drawn. Thus, several more independent experiments should be done or each experiment should be performed well enough in order to obtain useful information.

18. In Line 436-440, this statement is misleading as Rad52 also plays an important role in strand annealing of displaced nascent DNA strand with 3'-ssDNA of the resected second end in the HR pathway SDSA. Therefore, the effect of Rad52 inhibition is not limited to SSA-TI in the HR-proficient cells.

19. It would be better to further validate the point that BLM loss has more significant impact on SSA-mediated TI than on HR-mediated TI if the authors had analyzed the effect of BLM loss on SSA and HR induced by I-SceI and Cas9 using the DR-GFP and SSA-GFP reporter.

20. In Fig. 4d, the color columns for different Cas9 fusions are not clearly indicated. Also, as BLM deficiency enhances HR-mediated gene targeting, the title of this figure is inaccurate and should be reworded.

Responses to Reviewers' Comments

We sincerely appreciate the thoughtful comments and suggestions provided by the Referees and the Editor. We are now submitting our revised manuscript **"Rad51-independent gene targeting: synergistic enhancement by MSH2 loss, BLM loss, and targeted DNA breaks independent of cell cycle phase"**.

Reviewer #1 (Remarks to the Author):

Saito and Adachi show that targeted integration in mammalian cells, whether or not induced by DSB-generation using Cas9, can take place via a Rad52-mediated mechanism. This SSA-TI, as the authors call it, also takes place during the G1-phase of the cell-cycle and can still occur even if sequence homology between target site and template is not 100%.

I appreciate that the authors make an extensive effort to investigate the nature of Rad51-independent DNA integration. However, the finding that this is through a Rad52-dependent mechanism might not be very surprising, given the fact that this is the other homology-dependent recombination pathway present in mammalian cells. More importantly, some of the main conclusions, in particular concerning SSA-TI primarily taking place in G1, are not sufficiently supported by the data, as I will explain in more detail below.

Thank you very much for the favorable comments. As stated below, we have toned down statements throughout the manuscript on G1-phase preference of the Rad52-dependent mechanism.

Major comments:

1. One of the main conclusions is that a fraction of TI is Rad52-dependent. To support this, the authors show that treatment with two Rad52 inhibitors blocks TI in HR-deficient cells. However, both AICAR and DOPA are rather unspecific drugs that

for example activate AMPK (AICAR) and inhibit APE1 (DOPA). The authors validate the involvement of Rad52 with a Rad52 shRNA, but the knockdown observed with this shRNA is very poor (extended data 2a). To show conclusive evidence for this main conclusion, the authors should show a TI phenotype in cells that lost Rad52 expression using either a properly working shRNA or CRISPR-generated Rad52 KO cells.

We employed a **RAD52 knockout** cell line (which was created by conventional gene targeting without using CRISPR; we originally intended to publish this line in another paper) to reinforce our results observed with Rad52 inhibitors. As shown in Supplementary Fig. 6e, experiments using the *RAD52* knockout cell line gave essentially the same results as those obtained with Rad52 inhibitors. Therefore, we believe that despite their possible off-target actions, both AICAR and DOPA are reliable drugs in investigating the role for Rad52 in Rad51-independent gene targeting.

2. Furthermore, whereas the authors are at times careful to suggest that the Rad52-dependent TI is SSA, they do strongly suggest this at other times, and also refer to it as SSA-TI. To more firmly conclude that the Rad51-independent integration is in fact SSA, the authors should assess SSA-frequency upon loss of another SSA-factor, like XPF/ERCC1. This might require including a non-homologous region in the repair template, but this can easily be engineered and would allow to test the SSA-hypothesis.

Thank you for the suggestion. We have actually performed the **XPF/ERCC1 gene-knockdown experiments** and confirmed that SSA-TI is negatively affected when XPF/ERCC1 expression is significantly reduced, as shown in Supplementary Fig. 6g.

3. Based on the results presented in figure 3d-h and ext. data 8, the authors conclude that SSA is preferably active in G1 (lines 383-386). I think the data are insufficient to support this conclusion, based on the following:

a. In all SSA reporter assays, levels of SSA are comparable when using G1-Cas9 or

S/G2-Cas9 (fig. 3f, ext. data 8b), and even higher for S/G2-I-SceI than for G1-I-SceI (ext. data 8c).

b. Using TI frequency with the 95-95 template as a read-out for SSA, G1-Cas9 performs better than S/G2-Cas9 in some assays (Nalm-6 WT, HT1080), but not in others (Nalm-6 LIG4/POLQ^{-/-}). The data for the Nalm-6 RAD54^{-/-} cells are n=1 and therefore inconclusive.

c. The system is leaky, as there might still be some expression of G1-Cas9 in S/G2 (and the other way around), and a break generated in G1 might be repaired in S/G2.

d. The FUCCI-Cas9 system is not validated, so expression levels of the different variants might be incomparable and not as cell-cycle specific as suggested by the inferred data shown in panel 3d.

Thank you very much for the valuable comments. Our responses are as below.

4. To solve this, the authors should at least:

a. Assess the protein levels of the FUCCI-Cas9 variants throughout the different cell-cycle phases to validate their system.

We used different anti-Cas9 antibodies to assess the protein levels of the FUCCI-Cas9 variants in G1 and outside G1. As shown in Supplementary Fig. 9d, western blot analysis revealed that **S/G2-Cas9 is undetectable in G1, while G1-Cas9 is highly expressed in G1**. We noticed that S/G2-Cas9 expression levels are somewhat lower than G1-Cas9 levels, perhaps reflecting the duration or strength of Cdt1 or Geminin expression during the cell cycle. Despite this, we would like to emphasize the fact that expression of S/G2-Cas9, unlike G1-Cas9, led to significantly enhanced HR in gene-targeting assay as well as in DR-GFP assay (Fig. 3e).

b. Repeat the TI assays in the Nalm-6 RAD54^{-/-} cells to be able to make a statistical comparison between G1-Cas9 and S/G2-Cas9.

We repeated the TI assays (which is time and effort consuming because ~1,500 colonies should be picked and analyzed in each experiment) and confirmed

that G1-Cas9 is at least comparable to S/G2-Cas9 in enhancing SSA-TI.

c. Explain why some of the TI assays suggest that SSA might be more dominant in G1, while the SA-GFP reporter assays do not show this phenotype.

We appreciate this insightful comment. Although the exact reason for the different outcomes is currently unclear, we speculate that it may be attributed to the difference in the assay methods. For example, SSA-TI requires two SSA reactions (5' arm and the genome, and the genome and 3' arm), while the SA-GFP reporter only requires one SSA reaction between 100% homologous sequences. Given the low frequency of SSA occurring between homeologous sequences, G2-Cas9 may be more competent in enhancing chromosomal SSA (SA-GFP assay) than SSA-TI (TI assays).

d. Tone down the conclusion, as the results indicate that SSA might also occur in G1, but are not sufficient to conclude that SSA preferentially occurs in G1.

As suggested, we have toned down the conclusion regarding the G1 specificity throughout the text.

5. Although technically challenging, the authors could use inducible systems like iCas9 or UV-inducible guide RNAs in combination with cell-cycle synchronization methods to further validate G1-specific SSA.

Thank you very much for the suggestions. We agree that iCas9 or UV-inducible guide RNAs may be useful to further validate G1-specific SSA and specific factors involved. Although these issues are beyond the scope of this article, future efforts will be made in this direction.

6. Based on fig. 4a, b, the authors conclude that BLM-loss enhances SSA-TI more so than HR-TI. However, this cannot be concluded, as the 95-95 template can also integrate by HR-TI in the BLM-/- background, as shown in fig. 4c. Thus, the stronger fold-change in TI-frequency when using the 95-95 template is caused by a combined

effect on SSA-TI and HR-TI. To conclude that BLM loss affects SSA-TI more than HR-TI, the authors should compare TI frequency between WT and BLM^{-/-} cells that are treated with Rad51i or Rad52i.

In the revised manuscript, we have added a graph that directly compared TI frequency between WT and BLM^{-/-} cells treated with Rad51i or Rad52i (Supplementary Fig.10d).

7. The authors suggest that SSA-TI could be used for gene targeting in postmitotic cells or HR-deficient cells. The authors could show this to demonstrate the applicability of their findings. I would suggest to perform Cas9-mediated gene editing in HR-deficient cells like MDA-MB-436 or PEO-1 (eg by GFP-tagging an endogenous gene), and demonstrate that this is dependent on Rad52, and can be enhanced by BLM inhibition.

Thank you very much once again for the valuable suggestion. In the revised manuscript, we added our data using **MDA-MB-436 cells**, in which BRCA1 is mutated and hence HR activity is extremely low. Additional experiments indeed showed that gene targeting in MDA-MB-436 cells is dependent on Rad52, and can be enhanced by BLM inhibition (Supplementary Fig. 12).

Minor comments:

1. Are the PCR-products to assess TI in the puro- and 6TG-resistant colonies always exactly the predicted size? Imaginably, SSA-TI could be a bit more flexible and occur upon imperfect annealing resulting in deletions.

Yes, because SSA- as well as HR-mediated gene targeting is completely homology (or homeology) dependent, the PCR products to assess the event is always exactly the predicted size. SSA-TI is not that flexible (at least in the context of gene targeting), even though it allows sequence divergence.

2. Line 151-152: "This may imply that a lesser extent of end-resection is sufficient to

initiate SSA". This is an unlikely explanation, as most studies would suggest that SSA is more dependent on long range end-resection factors than HR (see for example van de Kooij et al. Nat Commun 13, 2022 and Tomimatsu et al., J Biol Chem 292, 2017).

We reworded the sentence to avoid the confusion.

[Confidential response to the Editor and Reviewers]

In SSA-mediated Alu recombination occurring in the chromosome, we know that Alu sequences closer to the DNA break are favored. This implies that extensive end-resection is not necessary for such SSA events, a phenomenon that is not in good accordance with the current view that SSA requires long-range end resection.

3. Line 171-173: "TMEJ does not have any role in HR-independent TI events". In fact, MMEJ-dependent Cas9-TI has been demonstrated to occur (Nakade S et al. Nat Commun 9, 3270 (2018)). Whereas loss of POLQ does not affect TI in Nalm-6 cells, this might be dependent on the repair template, or cell-line, used. The authors should comment on this.

We omitted the sentence. We agree that in certain gene editing technology TMEJ plays a role in HR-independent TI events when a target DSB is introduced. We appreciate the comment.

4. I am not sure what the value is of showing gene targeting efficiency. If I understand correctly, it looks at all integrations within the gene (because 6-TG resistant), which can be either targeted, or random. I would argue that showing the frequency of TI vs RI is sufficient, and whether or not there is also random integration that happens to occur within the gene is less relevant, and perhaps even confusing.

We respectfully think that although showing the frequency of TI vs RI may be sufficient in this paper, gene targeting efficiency is also important information and we prefer showing it because this is all about gene targeting. Additionally, as shown in Figure 1a, gene targeting efficiency is important to calculate TI and RI frequencies.

This is because in gene targeting assays, not all drug-resistant colonies are picked and subjected to further analysis.

5. The paper would benefit from reordering the figures to prevent going back and forth too much. For example, fig. 1e is discussed before 1d, extended data 2 before most of extended data 1.

In the revised version, we have made efforts to reorder several figures where possible or apparently beneficial. In the revised version, we changed Fig. 1d and Fig. 1e, and also Supplementary Figs 1 and 2.

Thank you very much once again for all of the useful comments.

Reviewer #2 (Remarks to the Author):

In this manuscript, Saito and Adachi present a very comprehensive and detailed study into the biology underlying gene targeting. They uncover a mechanism alternative to Rad51-guided HR that is able to target ectopically provided DNA to genomic loci. This SSA mechanism (termed SSA primarily because it is Rad52 dependent) is stimulated by DNA breaks at the locus and suppressed by the repair factors Msh2 and Blm. Of particular interest is the observation (and careful experimental design) that Cas9 fusion proteins that are expressed at specific cell cycle stages affect the SSA-mediated targeting differently. It is concluded that the mechanism (can) acts in G1, which would also be of potential clinical importance given the future genome editing may require the targeting of non-dividing cells.

All in all, I find the paper impressive, both on the depth, the quality of the experimental design and data, the clarity of the writing. I also find the novelty more than sufficient to warrant publication in a journal with broad readership (as is Nat. commun). I thus fully support publication of a manuscript that incorporates the underneath.

Thank you very much for the favorable comments.

The only “major” issue I have is whether the authors can be more convincing (perhaps I missed the argument or the experiment in support) that SSA-mediated targeted integration at break sites actually takes place in G1 (as opposed to being initiated in G1). They demonstrate that the frequency goes up when DSBs are generated in G1 but that doesn’t necessarily mean that SSA is happening in G1. I could also envisage a scenario where DSBs generated in G1 are processed such (for instance upon ensuing DNA replication when cells progress to S) that they have a different configuration compared to DSB that are generated in S/G2. I could think of very complex technically challenging experiments, which may (or may not) provide an answer to this question, but feel that this would be beyond what is reasonable to ask. However, if the authors are unable to provide evidence that SSA-TI is “finished” while cells are still in G1, I would find it more reasonable to i) adapt the text to accommodate this possibility (in abstract and throughout the text), and discuss this option in e.g. discussion section.

Thank you very much once again. As stated in our response to Reviewer 1, we have toned down all statements about G1-phase preference of the SSA-dependent mechanism. We would like to emphasize, however, that expression of S/G2-Cas9, unlike G1-Cas9, led to significantly enhanced HR in gene-targeting assay as well as in DR-GFP assay (Fig. 3e). If G1-Cas9 DSBs persist substantially to cause SSA in S/G2, **it would be unexplainable why HR-TI and HR repair (DR-GFP) is quite low when G1-Cas9 is expressed**, while S/G2-Cas9 greatly stimulates HR.

Minor issues:

i) I feel the title “Compelling evidence” is rather unconventional and also a somewhat unscientific/subjective. Merely removing “Compelling” doesn’t do the trick. One would hope that all papers have “evidence” for what is presented.

Thank you for the advice. We have changed the title to:
"Rad51-independent gene targeting: synergistic enhancement by MSH2 loss, BLM loss, and targeted DNA breaks independent of cell cycle phase".

ii) Please specify the Extended figure panels where appropriate. The figures are quite extensive and it would help the reader to point to the specifics: For instance in line 110, refer to Extended Data Fig. 1A+B, as opposed to only Fig. 1), and so forth.

Thank you for the advice. We made several modifications in the revised manuscript to make the figure panels more reader-friendly.

iii) Line 142, I could not find the 1.5- fold (in Ex Fig 1F).

We reworded the sentence including the 1.5-fold, which was indeed inappropriate.

iv) Line 151-152 holds unnecessary speculation, which I don't agree with: to me, the mechanism for HR is much more complex than SSA, as it needs DNA synthesis on a template (first processing of the flap prior to DNA synthesis) and therefore there are many potential reasons for why SSA is more rapid, not at all pointing to end-resection (kinetics).

We deleted the unnecessary speculation and reworded the sentence.

v) Line 162: wouldn't 1.5-2 fold be more appropriate (2 fold is somewhat overly optimistic given the data presented in Ext Figs 1E and 1C).

Again, we agree that 1.5-2 fold is more appropriate for interpretation and expression of the data. We appreciate the advice.

I would like to end with complimenting the authors on this impressive piece of work (impressive also considering that just one person carried out all experiments).

Thank you very much once again for all the favorable comments and the compliment.

Reviewer #3 (Remarks to the Author):

In this manuscript, Saito & Adachi used gene targeting to the HPRT gene, together with the DR-GFP and SA-GFP reporters, to analyze the role of SSA in homology-directed gene targeting. The question asked by the authors has been asked and addressed before, but this study presents some new findings: 1) Rad52 inhibition suppresses gene targeting and homology-directed repair (HDR) in HR-deficient cells, not in HR-proficient cells, suggesting involvement of SSA; 2) This SSA mechanism tolerates 5% divergence in homology arms but HR does not, thus giving Msh2 more suppressive control in SSA; 3) SSA-mediated targeted and random integration also occurs in G1 in addition to S/G2; 4) BLM loss has more significant impact on SSA-mediated TI than on HR-mediated TI. These findings are very interesting. In particular, G1 occurrence of SSA may help clarify the view on the cell cycle stage for SSA although the exact regulation is yet to be determined. In this regard, the evidence is convincing to me but additional experiments are needed to solidify this point as explained below. The authors also stated that this study has implications for precise gene editing in post-mitotic cells. However, the level of SSA-mediated TI is quite low, limiting practical use of this strategy in precise gene editing and by extension the significance of this manuscript. Additional comments are also detailed below.

Thank you very much for the favorable comments. We have made efforts to solidify the points suggested. We agree that practical use of the SSA-TI strategy needs significant improvements, our paper can be the first small step towards achieving precise gene editing in post-mitotic cells, because we revealed that NHEJ/TMEJ suppression, BLM inhibition, and DSB induction all enhance SSA-TI frequency.

Detailed comments:

1. The title of this manuscript is vague and does not truly reflect the key findings of this study.

Thank you for the comment. We have changed the title to:
"Rad51-independent gene targeting: synergistic enhancement by MSH2 loss, BLM loss, and targeted DNA breaks independent of cell cycle phase".

2. The efficiency of SSA-mediated TI is much lower by up to 25-fold than Rad51-mediated TI. Thus, applications of SSA-mediated TI in genome editing could be very limited to non-growing cells and insignificant in proliferating cells. In addition, increasing the efficiency of SSA-mediated TI is still needed for practical use even in non-growing cells. It would strengthen the manuscript if this study had tested SSA-mediated TI by quantitative PCR and sequencing in slow-growing cells or even non-growing cells.

In the revised manuscript, we have toned down the sentences for practical use of SSA-TI; however, we have performed additional experiments using the **HR-deficient MDA-MB-436 cell line** whose doubling-time is quite long (~45 hr), and demonstrated that SSA-TI gene targeting is indeed feasible in such slow-growing cells (Supplementary Fig. 12).

3. The targeting vector contains a promoter-less 2A-Puro cassette. For RI, the cells would survive with puromycin selection as long as the 2A-puro is inserted into coding regions of genes within a correct reading frame. If this happens, the level of RI could be greatly underestimated in this assay.

As pointed out, targeting vectors containing a promoter-less 2A-drug cassette confers underestimated RI frequency (Saito, S. *et al.* FEBS J. 284:2748-2763, 2017). In this study, we focused on precise evaluation of TI frequency: we did an extensive number of gene targeting assays and picked an extensive number of drug-resistant colonies for further analysis; therefore, to minimize random integrants (i.e., non-TI drug^r colonies), it is reasonable to employ promoter-less vectors, which enabled us to calculate TI frequency in many different gene-targeting assays.

4. In this manuscript, it is unclear how many copies of the DR-GFP and SSA-GFP reporter stably integrated in cells. The authors need to quantify the copy number of

these reporters randomly integrated and restrict copies of the DR-GFP and SSA-GFP reporter in cells in order to avoid the undesired interference of homologous copies with SSA and HR.

The DR-GFP and SSA-GFP reporter substrate is stably integrated by conventional gene targeting into the *HPRT* locus (at exon3) of all cell lines tested. The correct gene-targeting event has been confirmed by PCR analysis, which shows that only one copy is inserted at the targeted locus in each cell line.

5. The authors used only one site for targeted integration. If this site has some particular nucleotide composition and chromatin context, which could affect the control of targeted integration, it is possible that analysis in this manuscript could not be generalized. I would suggest additional validation at one more site away from the site used already in the manuscript.

We appreciate this suggestion. We performed additional experiments targeting exon 6 of the *HPRT* gene (Supplementary Fig. 11), which led us to confirm that SSA-mediated TI is indeed a general phenomenon.

6. The authors used a few KO cell lines in this study but did not provide details how these cell lines or particular clones were generated, confirmed and chosen.

We have shown more clearly how the KO cell lines were obtained or created in the Methods section and Supplementary Figures, citing relevant papers more properly.

7. In Extended Data Fig. 2a, c, Rad52 shRNA used is not that effective. Is it a reason why there exists inconsistency in the effect on gene targeting efficiency between Rad52i treatment and Rad52 shRNA in Extended Data Fig. 2b, d?

We agree that Rad52 shRNA used was not effective. In the revised version, we additionally employed a **RAD52 knockout cell line** (which was created by conventional gene targeting without CRISPR; we originally intended to publish this

line in another paper). As shown in Supplementary Fig. 6e, experiments using the *RAD52* knockout cell line gave essentially the same results as those obtained with Rad52 inhibitors.

8. Fig. 1e is presented again in Extended Data Fig. 2d.

Yes, we did that intentionally just to make the paper reader friendly. The fact that the same figure is presented again has been mentioned in the figure legend.

9. In Line 71-74, “100% efficiency of gene targeting” is misleading and should be reworded.

10. In Line 85-87, “less than 5% of Rad54/Rad54B-proficient cells” is confusing and should be reworded as “less than 5% of the frequency in Rad54/Rad54B-proficient cells”.

11. In Line 171-173, the statement “It should be noted, however, that TMEJ, unlike SSA (which requires a longer sequence of homology than TMEJ), does not have any role in HR-independent TI events” is not accurate. In the gene targeting assays, the TI frequency was slightly higher in LIG4 KO/POLQ KO cells treated with Rad51 inhibitors, but the TI levels were not determined in POLQ KO cells. Therefore, this conclusion could not be drawn.

Thank you very much for all of these comments. We have reworded all these sentences.

12. In Extended Data Fig. 7a, site-specific DSB induction in the genome promotes RI at other genomic regions. Why?

We speculate that this RI enhancement is due to both on-target RI (at the *HPRT* locus) and off-target RI events (at off-target DSB site).

13. In Extended Data Fig. 7d, as TI with short homology arms (80 bp and 212 bp) is mediated primarily by Rad52-dependend SSA and suppressed by Msh2, it is

expected that 5% divergence in short homology arms would not affect TI with short homology arms (80 bp and 212 bp). But is it really the case? No data were provided.

Although we are ready to remove the data of the short homology arm vectors, which can only cause TI in the absence of LIG4 and POLQ, we prefer to present the data as they are. To decipher the details on how these vectors are affected by sequence divergence, considerable efforts exceeding the scope of this study are needed, and we think that it should be pursued as a separate comprehensive study on SSA and mismatch repair factors.

14. The authors used the Fucci system to control the Cas9 protein level in specific cell cycle phase and intended to induce site-specific DSB at a cell cycle phase-specific manner. However, as demonstrated previously, Cas9 can remain tightly bound to cleaved target for a long period of time (up to 6-7 hours), affecting DNA repair (Stenberg et al, Nature, 2014, 506:62-67; Richardson et al, Nat Biotech, 2016, 34:339-344; Liu et al, Genome Biol, 2022, 23:165). Thus, DSBs induced by Cas9 in the G1 phase could be exposed only later in the S phase for DNA repair, interfering with the analysis in this study. In addition, although the Cas9 protein level is controlled by hCdt1 degradation in a cell cycle-specific manner, it is unclear whether target-bound Cas9 can be degraded before being released from its targets. It is likely that in order to degrade target-bound Cas9 in proteosomes, release from its targets in the context of chromatin and getting into the cytoplasm from the nucleus are necessary. In addition, it is unclear whether Cas9 cleaves DNA in similar efficiencies between different cell cycle stages. Therefore, the authors should provide evidence in this regard by analyzing both the cell cycle distribution of Cas9 protein and the efficiency of Cas9-induced DNA cutting in different cell cycle phases.

Thank you very much for this important comment. We performed western blot analysis and showed in the revised manuscript that **S/G2-Cas9 is not detectable in the G1 phase, while G1-Cas9 is hardly detectable in the S-G2 phases** (Supplementary Fig. 9d). Although the exact nature and regulation of Cas9-induced DNA cutting in different cell cycle phases awaits further studies, we would like to emphasize that we showed clearly in this study that **HR is stimulated**

by S/G2-Cas9 and, as expected, not by G1-Cas9 (Fig. 3e). This was observed in the DR-GFP assay, and not in the SA-GFP assay. If G1-Cas9 possessed some unexplainable unknown characteristics, these results would not have been obtained.

15. In Line 383, the authors stated: “Surprisingly, G1-Cas9 expression showed a greater effect in enhancing SSA than did S/G2-Cas9 expression, suggesting that SSA-mediated DSB repair operates in G1, rather than outside G1 phase (Fig. 3f and Extended Data Fig. 8b).” Given a considerable level of SSA outside the G1 phase, this statement is inaccurate.

16. In Line 410-411, the statement “Taken together, we conclude that SSA-TI primarily occurs in the G1 phase of the cell cycle” is not accurate as SSA and SSA-TI are robust and comparable in both G1 and S/G2 (Fig. 3f, g and Extended Data Fig. 8).

We have reworded the sentences to tone down the conclusion regarding G1 preference of SSA-TI.

17. In many cases (e.g., Fig. 3g, Fig. 4d and Extended Data Fig. 2d), the difference in results between three independent experiments is too large, resulting in large standard deviation, any conclusion determined by statistical analysis with one way or another could be misleading unless no conclusion is drawn. Thus, several more independent experiments should be done or each experiment should be performed well enough in order to obtain useful information.

We highly appreciate these comments. We have removed unnecessary asterisks determined by statistical analysis that do not lead to any informative conclusion. Regarding Fig. 3g, we repeated experiments and obtained clearer results.

18. In Line 436-440, this statement is misleading as Rad52 also plays an important role in strand annealing of displaced nascent DNA strand with 3'-ssDNA of the resected second end in the HR pathway SDSA. Therefore, the effect of Rad52 inhibition is not limited to SSA-TI in the HR-proficient cells.

We have omitted this misleading statement.

19. It would be better to further validate the point that BLM loss has more significant impact on SSA-mediated TI than on HR-mediated TI if the authors had analyzed the effect of BLM loss on SSA and HR induced by I-SceI and Cas9 using the DR-GFP and SSA-GFP reporter.

We have performed all these experiments and obtained reasonable results (i.e., BLM loss stimulates SSA rather than HR), which are added in the revised manuscript (Supplementary Fig. 10c,g).

20. In Fig. 4d, the color columns for different Cas9 fusions are not clearly indicated. Also, as BLM deficiency enhances HR-mediated gene targeting, the title of this figure is inaccurate and should be reworded.

We have corrected the title of Figure 4, and clearly indicated the color columns in Fig. 4d. Thank you very much once again for all of the thoughtful comments.

REVIEWERS' COMMENTS

Reviewer #1 (Remarks to the Author):

The authors have appropriately addressed all of my concerns. The manuscript would be good for publication as is (in my opinion), but I would nevertheless highly recommend to make a few minor adjustments to tone down the conclusion that loss of BLM enhances SSA-TI more so than HR-TI (fig. 4 and sup fig 10). It is true that BLM loss enhances TI-frequency of the 95-95 vector with about 20-fold, whereas it enhances TI frequency of the 100-100 vector with only about 3-fold. However, in absence of BLM, TI of the 95-95 vector is not a bona fide read-out for SSA-TI anymore, as the authors also conclude themselves (lines 430-431 and 444-446). Therefore, the authors did the experiment shown in sup fig 10d, which clearly shows that the 20-fold increase in TI with the 95-95 vector is composed of an ~10-fold increase in HR-TI (i.e. Rad51-dependent TI) and an about 10-fold increase in SSA-TI (i.e. Rad52-dependent). Similarly, the 3-fold increase in TI of the 100-100 vector is a combined 1.5x HR-TI and 1.5x SSA-TI. Hence, in the TI assays, I would argue that BLM loss enhances HR and SSA equally. This is not in conflict with any of the main findings, so I don't think changing this specific conclusion would majorly affect the main message of the manuscript, whereas it would be a more correct interpretation of the data.

I also think I spotted a small error in Fig. S10C: In the SA-GFP panel, it is stated that the green bars represent RAD52^{-/-} cells, but this should be BLM^{-/-} cells (I suppose).

Reviewer #2 (Remarks to the Author):

The authors carefully and to my satisfaction addressed the issues I raised concerning their initial submission. I would thus support publication of this study/manuscript.

Reviewer #3 (Remarks to the Author):

In this revision, some of the issues have been adequately addressed. However, I still have some reservations on the current form.

1. In the revised manuscript, Supplementary Fig. 9d demonstrates that G1-Cas9 is detectable in the S-G2 phase although the level is low. The G1-Cas9 protein detectable in the S-G2 phase might be the Cas9 protein that cleaves DNA target and is tightly bound to cleaved target in the G1 phase and disassociated from DNA in the S-G2 phase (Stenberg et al, Nature, 2014, 506:62-67; Richardson et al, Nat Biotech, 2016, 34:339-344; Liu et al, Genome Biol, 2022, 23:165). As a result, SSA repair of DSBs induced by G1-Cas9 could take place in the S-G2 phase. Given a recent study showing that even a small amount of SpCas9-sgRNA in the cells is sufficient to induce DSB repair at the level comparable to that by 10-25-fold more SpCas9-sgRNA (Liu et al, Genome Biol, 2022, 23:165), it is possible that a significant level of G1-Cas9-induced DSBs are repaired by SSA in the S-G2 phase. In the revised manuscript, it is unclear to what extent SSA induced by G1-Cas9 occur in the S-G2 phase. The manuscript appears to suggest that SSA induced by G1-Cas9 only occurs in the G1 phase. In this regard, I am not fully convinced.

2. To address the issue regarding the copy number of the reporters randomly integrated in the cells in addition to the reporters targeted into the locus as intended, the authors used PCR to confirm that only one copy is inserted at the targeted locus. However, the question about random integration of the reporters is left unanswered.

3. The revised manuscript is challenging to follow, particularly in the Methods section. It is good to have details with sufficient information but should be more concise and coherent.

Responses to Reviewers' Comments

We sincerely appreciate the thoughtful comments and suggestions provided by the Referees and the Editor. We are now submitting our revised manuscript with a new title: **"Characterization and regulation of cell cycle-independent noncanonical gene targeting"**.

Reviewer #1 (Remarks to the Author):

The authors have appropriately addressed all of my concerns. The manuscript would be good for publication as is (in my opinion), but I would nevertheless highly recommend to make a few minor adjustments to tone down the conclusion that loss of BLM enhances SSA-TI more so than HR-TI (fig. 4 and sup fig 10). It is true that BLM loss enhances TI-frequency of the 95-95 vector with about 20-fold, whereas it enhances TI frequency of the 100-100 vector with only about 3-fold. However, in absence of BLM, TI of the 95-95 vector is not a bona fide read-out for SSA-TI anymore, as the authors also conclude themselves (lines 430-431 and 444-446). Therefore, the authors did the experiment shown in sup fig 10d, which clearly shows that the 20-fold increase in TI with the 95-95 vector is composed of an ~10-fold increase in HR-TI (i.e. Rad51-dependent TI) and an about 10-fold increase in SSA-TI (i.e. Rad52-dependent). Similarly, the 3-fold increase in TI of the 100-100 vector is a combined 1.5x HR-TI and 1.5x SSA-TI. Hence, in the TI assays, I would argue that BLM loss enhances HR and SSA equally. This is not in conflict with any of the main findings, so I don't think changing this specific conclusion would majorly affect the main message of the manuscript, whereas it would be a more correct interpretation of the data.

We really appreciate this insightful comment. Although the major conclusions of this paper are not affected, we agree that interpretation of BLM results has room for improvement. For this reason, we have slightly modified the main text to state more clearly that BLM loss enhances SSA as well as HR.

I also think I spotted a small error in Fig. S10C: In the SA-GFP panel, it is stated that the green bars represent RAD52^{-/-} cells, but this should be BLM^{-/-} cells (I suppose).

We now have corrected the error in Supplementary Fig. 10C. Thank you very much once again for all of the useful comments.

Reviewer #2 (Remarks to the Author):

The authors carefully and to my satisfaction addressed the issues I raised concerning their initial submission. I would thus support publication of this study/manuscript.

Thank you very much once again for the favorable comments.

Reviewer #3 (Remarks to the Author):

In this revision, some of the issues have been adequately addressed. However, I still have some reservations on the current form.

1. In the revised manuscript, Supplementary Fig. 9d demonstrates that G1-Cas9 is detectable in the S-G2 phase although the level is low. The G1-Cas9 protein detectable in the S-G2 phase might be the Cas9 protein that cleaves DNA target and is tightly bound to cleaved target in the G1 phase and dissociated from DNA in the S-G2 phase (Stenberg et al, Nature, 2014, 506:62-67; Richardson et al, Nat Biotech, 2016, 34:339-344; Liu et al, Genome Biol, 2022, 23:165). As a result, SSA repair of DSBs induced by G1-Cas9 could take place in the S-G2 phase. Given a recent study showing that even a small amount of SpCas9-sgRNA in the cells is sufficient to induce DSB repair at the level comparable to that by 10-25-fold more SpCas9-sgRNA (Liu et al, Genome Biol, 2022, 23:165), it is possible that a significant level of G1-Cas9-induced DSBs are repaired by SSA in the S-G2 phase. In the revised manuscript, it is unclear to what extent SSA induced by G1-Cas9 occur in the S-G2 phase. The manuscript appears to suggest that SSA induced by G1-Cas9 only occurs in the G1 phase. In this regard, I am not fully convinced.

Thank you very much once again for this important comment. We agree with the view that the G1-Cas9 protein slightly detectable in the S-G2 phase might have contributed to SSA. We believe that the reviewer's concern should be solved by the following two points:

1) Although the exact nature and regulation of Cas9-induced DNA cutting in different cell cycle phases awaits further studies, again we respectfully emphasize that we have shown using reporter assays that HR is significantly stimulated by S/G2-Cas9 and not by G1-Cas9. If G1-Cas9 possessed some unexplainable or unknown characteristics, these results would not have been obtained (more specifically, HR should have been enhanced by G1-Cas9 in the DR-GFP assay).

2) Even more importantly, similar results have been obtained using I-SceI endonuclease fused to hCdt1 or hGeminin, as has been stated in the text and shown in Supplementary Fig. 9g. Thus, SSA enhancement in G1 phase is not confined to DSBs induced by Cas9, which is somewhat unique as the reviewer has been suggesting.

In the revised manuscript, we have added the following sentences to indicate these points, citing the suggested three papers:

"One might think that G1 occurrence of SSA needs to be interpreted with caution because Cas9-induced DSBs are unique in that they are RNA-mediated; in addition, G1-Cas9 protein could possibly persist at the break site until the S-G2 phase to cause SSA (Stenberg et al, Nature, 2014, 506:62-67; Richardson et al, Nat Biotech, 2016, 34:339-344; Liu et al, Genome Biol, 2022, 23:165). In this respect, it is important to mention that G1-Cas9 expression is highly inefficient in enhancing HR (Fig. 3e and Supplementary Fig. 9e). Moreover, we obtained essentially the same results with I-SceI endonuclease; specifically, I-SceI-hCdt1 does enhance SSA similarly to I-SceI-hGeminin (Supplementary Fig. 9g). Thus, these results suggest that SSA enhancement upon G1-phase DSBs is a general phenomenon, not confined to Cas9-induced DSBs."

2. To address the issue regarding the copy number of the reporters randomly integrated in the cells in addition to the reporters targeted into the locus as intended, the authors used PCR to confirm that only one copy is inserted at the targeted locus.

However, the question about random integration of the reporters is left unanswered.

Because knocking-in of the GFP reporter substrate was performed by conventional gene targeting (without using Cas9), the copy number is 1. This is crystal clear because TI and RI do not occur simultaneously in the same clone (TI frequency: 2×10^{-6} vs RI frequency: 1×10^{-6} ; hence, random integration occurrence is highly unlikely in the targeted clones). Additionally (this may not be necessary, but just in case), the correct gene-targeting event has been confirmed by PCR analysis, which shows that only one copy is inserted at the targeted locus (and not at random sites) in each cell line. Also, we isolate more than 2 targeted clones, and at least two of these are subjected to preliminary experiments, further supporting our statements.

3. The revised manuscript is challenging to follow, particularly in the Methods section. It is good to have details with sufficient information but should be more concise and coherent.

Thank you very much for the suggestions. In the revised version, we have made efforts to make the paper more reader friendly, especially in the Methods section where several contents are reorganized or moved to the Supplementary Methods. Also, we have provided Supplementary Table 5, as a list of the antibodies used, to make the Methods section concise. Thank you very much once again for all of the thoughtful comments.